# Joint cell segmentation and cell type annotation for spatial transcriptomics

Russell Littman[1,2,3], Zachary Hemminger[2,4], Robert Foreman[2], Douglas Arneson[2,5], Guanglin Zhang[1], Fernando Gómez-Pinilla[1], Xia Yang[1,2,3,*,†] (iD) & Roy Wollman[1,2,3,4,**,†] (iD)

## Abstract

**RNA hybridization-based spatial transcriptomics provides unparalleled detection sensitivity. However, inaccuracies in segmentation of image volumes into cells cause misassignment of mRNAs which is a major source of errors. Here, we develop JSTA, a computational framework for joint cell segmentation and cell type annotation that utilizes prior knowledge of cell type-specific gene expression. Simulation results show that leveraging existing cell type taxonomy increases RNA assignment accuracy by more than 45%. Using JSTA, we were able to classify cells in the mouse hippocampus into 133 (sub)types revealing the spatial organization of CA1, CA3, and Sst neuron subtypes. Analysis of within cell subtype spatial differential gene expression of 80 candidate genes identified 63 with statistically significant spatial differential gene expression across 61 (sub)types. Overall, our work demonstrates that known cell type expression patterns can be leveraged to improve the accuracy of RNA hybridization-based spatial transcriptomics while providing highly granular cell (sub)type information. The large number of newly discovered spatial gene expression patterns substantiates the need for accurate spatial transcriptomic measurements that can provide information beyond cell (sub)type labels.**

**Keywords** cell segmentation and annotation; scRNAseq; single cell multiomics integration; spatial differentially expressed genes; spatial transcriptomics
**Subject Categories** Chromatin, Transcription & Genomics; Computational Biology; Methods & Resources
**Mol Syst Biol. (2021) 17: e10108**

## Introduction

Spatial transcriptomics has been employed to explore the spatial and cell type-specific gene expression to better understand physiology and disease (Asp *et al*, 2019; Burgess, 2019). Compared to other spatial transcriptomics methods, RNA hybridization-based approaches provided the highest RNA detection accuracies with capture rates > 95% (Lubeck *et al*, 2014). With the development of combinatorial approaches for RNA hybridization, the ability to measure the expression of hundreds to thousands of genes makes hybridization-based methods an attractive platform for spatial transcriptomics (Beucher, 1979; Najman & Schmitt, 1994; Al-Kofahi *et al*, 2010; Lubeck *et al*, 2014; Chen *et al*, 2015; Eng *et al*, 2019; preprint: Park *et al*, 2019; Vu *et al*, 2019; preprint: Petukhov *et al*, 2020; Qian *et al*, 2020; Yuste *et al*, 2020). Nonetheless, unlike dissociative approaches, such as single-cell RNA sequencing (scRNAseq) where cells are captured individually, RNA hybridization-based approaches have no a priori information of which cell a measured RNA molecule belongs to. Segmentation of image volumes into cells is therefore required to convert RNA detection into spatial single-cell data. Assigning mRNA to cells remains a challenging problem that can substantially compromise the overall accuracy of combinatorial FISH approaches.

Generation of spatial single-cell data from imaging-based spatial transcriptomics relies on algorithmic segmentation of images into cells. Current combinatorial FISH work uses watershed-based algorithms with nuclei as seeds, and the total mRNA density to establish cell borders (Najman & Schmitt, 1994; Chen *et al*, 2015; Eng *et al*, 2019). Watershed algorithm was proposed more than 40 years ago (Beucher, 1979), and newer segmentation algorithms that utilize state of the art machine learning approaches have been shown to improve upon classical watershed approach (Al-Kofahi *et al*, 2010; Vu *et al*, 2019). However, their performance is inherently bounded by the quality of the "ground truth" dataset used for training. In tissue regions with dense cell distributions, there is simply not enough information in the images to perform accurate manual labeling and create a sufficiently accurate ground truth training datasets. Therefore, there is an urgent need for new approaches that can combine image information with external datasets to improve image segmentation and thereby the overall accuracy of spatial transcriptomics.

1 Department of Integrative Biology and Physiology, UCLA, Los Angeles, CA, USA
2 Institute of Quantitative and Computational Biosciences, UCLA, Los Angeles, CA, USA
3 Bioinformatics Interdepartmental Program, UCLA, Los Angeles, CA, USA
4 Department of Chemistry and Biochemistry, UCLA, Los Angeles, CA, USA
5 Bakar Institute of Computational Health Sciences, UCSF, Los Angeles, CA, USA
*Corresponding author. Tel: +310 825 1812; E-mail: xyang123@ucla.edu
**Corresponding author. Tel: +855 810 0905; E-mail: rwollman@ucla.edu
†These authors contributed equally to this work as senior authors

Due to the deficiency in existing image segmentation algorithms, a few segmentation-free spatial transcriptomic approaches were proposed. pciSeq's primary goal is to assign cell types to nuclei by using proximity to mRNA, and an initialized segmentation map to compute the likelihood of each cell type (Qian *et al*, 2020). Similarly, SSAM creates cell type maps based on RNA distributions, without creating a cell segmentation map because it ignores cellular boundaries (preprint: Park *et al*, 2019). Therefore, while both pciSeq and SSAM leverage cell type catalogs to provide insights into the spatial distribution of different cell types, they do not produce a high-quality cell segmentation map. More recently, an approach for updating cell boundaries in spatial transcriptomics data has been developed (preprint: Petukhov *et al*, 2020). Baysor uses neighborhood composition vectors and Markov random fields to segment spatial transcriptomics data and identify cell type clusters.

Here, we present JSTA, a computational framework for jointly determining cell (sub)types and assigning mRNAs to cells by leveraging previously defined cell types through scRNAseq. Our approach relies on maximizing the internal consistency of pixel assignment into cells to match known expression patterns. We compared JSTA to watershed in assigning mRNAs to cells through simulation studies to evaluate their accuracy. Application of JSTA to MERFISH measurements of gene expression in the mouse hippocampus together with Neocortical Cell Type Taxonomy (NCTT) (Yuste *et al,* 2020) provides a highly granular map of cell (sub)type spatial organization and identified many spatially differentially expressed genes (spDEGs) within these (sub)types (Lein *et al,* 2007).

## Results

### JSTA overview and method

Our computational framework of JSTA is based on improving initial watershed segmentation by incorporating cell (sub)type probabilities for each pixel and iteratively adjusting the assignment of boundary pixels based on those probabilities (Fig 1A).

To evaluate JSTA, we chose to use the mouse hippocampus for two reasons: (i) The mouse hippocampus has high cell (sub)type diversity as it includes more than 35% of all cell (sub)types defined by the NCTT. (ii) The mouse hippocampus has areas of high and low cell density. These two reasons make the mouse hippocampus a good test case for the hypothesis that external cell (sub)type-specific expression data could be leveraged to increase the accuracy of spatial transcriptomics, as implemented in our approach. We performed multiplexed error robust fluorescent in situ hybridization (MERFISH) of 163 genes which include 83 selected cell marker genes, which show distinct expression between cell types and are used for cell classification and segmentation and 80 genes previously implicated with biological importance in traumatic brain injury (Fig 1B). Combining this MERFISH dataset, DAPI stained nuclei, and the NCTT reference dataset using JSTA, we created a segmentation map that assigns all mRNAs to cells while simultaneously classifying all cells into granular (sub)types based on NCTT.

In JSTA, we leverage the NCTT information to infer probabilities at the pixel level. However, learning these probabilities from NCTT is challenging for two reasons. (i) NCTT data were acquired with scRNAseq technology that has higher sparsity due to low capture rates and needs to be harmonized. (ii) NCTT data provide expression patterns at the cell level and not the pixel level. We expect the mean expression among all pixels in a cell to be the same as that of the whole cell. Yet, variance and potentially higher distribution moments of the pixel-level distribution are likely different from those of the cell-level distribution due to sampling and biological factors such as variability in subcellular localization of mRNA molecules (Eng *et al,* 2019). To address these issues, JSTA learns the pixel-level cell (sub)type probabilities using two distinct deep neural network (DNN) classifiers, a cell-level type classifier, and a pixel type classifier. Overall, JSTA learns three distinct layers of information: segmentation map, pixel-level classifier, and cell-level classifier.

Learning of model parameters is done using a combination of NCTT and the MERFISH data. The cell type classifier is learned directly from NCTT data after harmonization. The other two layers are learned iteratively using expectation maximization (EM) approach (Chen *et al,* 2015). Given the current cell type assignment to cells, we train a pixel-level DNN classifier to output the cell (sub)type probability of each pixel. JSTA can be applied on any user-selected subset of the genes; the local mRNA density of these selected genes around each pixel is used as the input for the pixel-level classifier. The selection of genes drives how well the cell type classifier can distinguish between distinct cell types. The updated pixel classifier is used to assign probabilities to all border pixels. The new probabilities are then used to "flip" border pixels' assignment based on their type probabilities. The updating of the segmentation map requires an update of the cell-level type classification which triggers a need for an update of pixel-level classifier training. This process is then repeated until convergence. Analysis of the mean pixel-level cell (sub)type classification accuracy shows an increase in the algorithm's classification confidence over time demonstrating that the NCTT external information gets iteratively incorporated into the tasks of cell segmentation and type annotation (Fig EV1). For computational efficiency, we iterate between training, reassignment, and reclassification in variable rates. As this approach uses cell type information to improve border assignment between neighboring cells, in cases where two neighboring cells are of the same type, the border between them will stay the same as the initial watershed segmentation. The final result is a cell type segmentation map that is initialized based on watershed and adjusted to allow pixels to be assigned to cells to maximize consistency between local RNA density and cell type expression priors.

### Performance evaluations

#### *Performance evaluation using simulated hippocampus data*

To test the performance of our approach, we utilized synthetic data generated based on the NCTT (Lein *et al,* 2007) (Fig 2A and B). Details on the synthetic generation of cell position, morphologies, type, and expression profiles are available in the Materials and Methods section. Using this synthetic data, we evaluated the performance of JSTA in comparison with watershed at different cell type granularities. For example, two cells next to each other that are of subtypes CA1sp1 and CA1sp4 would add to the error in segmentation, but if the cell type resolution decreases to CA1 cells, these

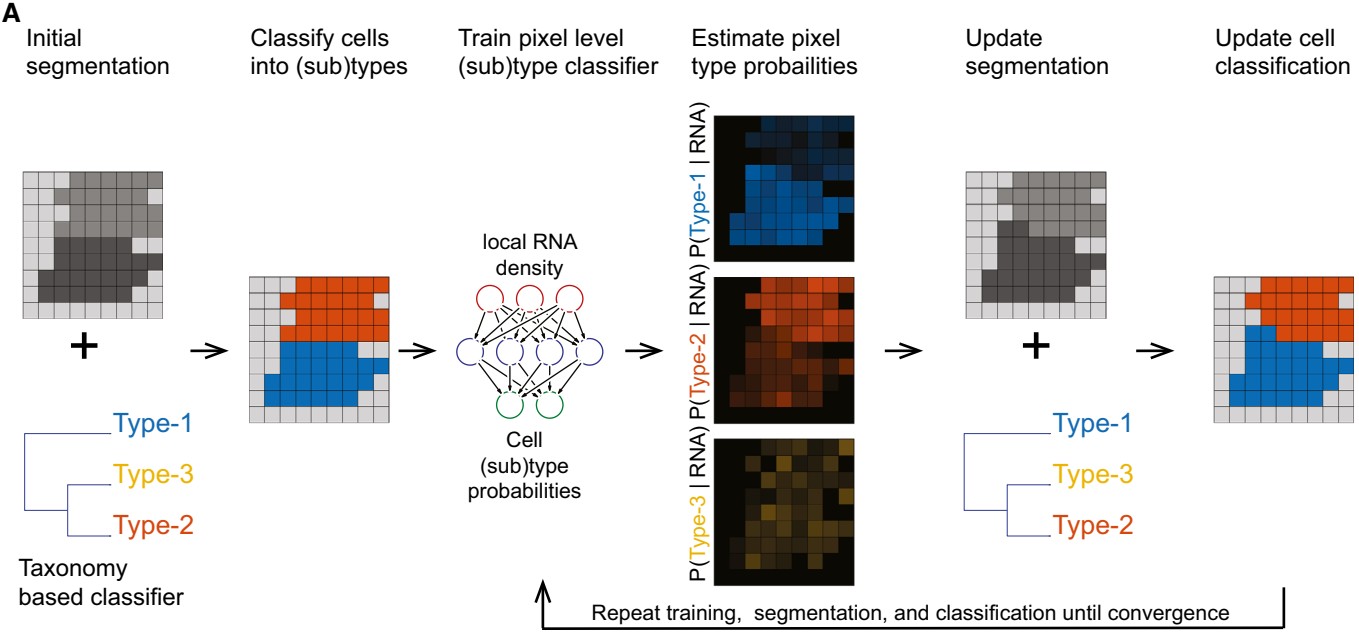

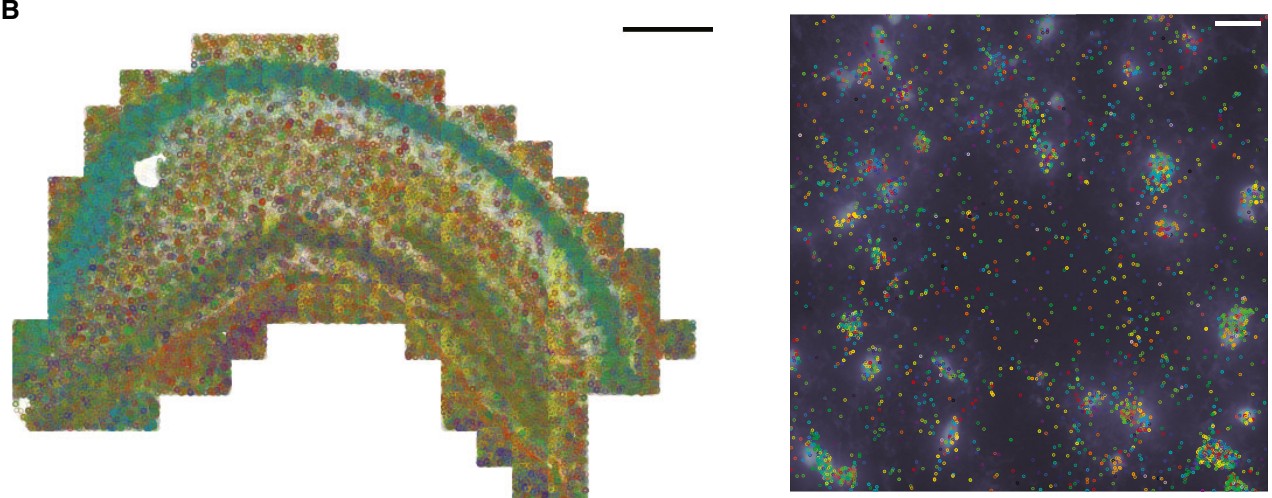

**Figure 1. Overview of JSTA and the spatial transcriptomic data used for performance evaluation.**

A   Joint cell segmentation and cell type annotation (JSTA) overview. Initially, watershed-based segmentation is performed and a cell-level type classifier is trained based on the Neocortical Cell Type Taxonomy (NCTT) data. The deep neural network (DNN) parameterized cell-level classifier then assigns cell (sub)types (red and blue in this cartoon example). Based on the current assignment of pixels to cell (sub)types, a new DNN is trained to estimate the probabilities that each pixel comes from each of the possible (sub)types given the local RNA density at each pixel. In this example, two pixels that were initially assigned to the "red" cells got higher probability to be of a blue type. Since the neighbor cell is of type "blue", they were reassigned to that cell during segmentation update. Using the updated segmentation and the cell type classifier cell types are reassigned. The tasks of training, segmentation, and classification are repeated over many iterations until convergence.

B   Multiplexed error robust fluorescent in situ hybridization (MERFISH) and DAPI stained nuclei in the mouse hippocampus. Each gene is represented by a different color. For the entire hippocampus (left), only the mRNA spots are shown with a scale bar of 500 μm. On the zoomed-in section (right), each gene is represented by a different color dot, and the DAPI intensity is displayed in white. The scale bar is 20 μm.

would be considered the same type, and misassignment of mRNA between these cells is no longer penalized. Evaluating the methods in this manner allows us to explore the trade-off between cell type granularity and mRNA assignment accuracy. Our analysis shows that JSTA consistently outperforms watershed at assigning spots to cells (Fig 2C). Interestingly, the benefit of JSTA was evident even with a small number of genes (Fig 2D). With just 12 genes, the performance jumps to 0.50 at the highest cell type granularity, which is already higher than watershed's accuracy; at a granularity of 16 cell types, the accuracy reached 0.62 (Fig 2C and D). Overall

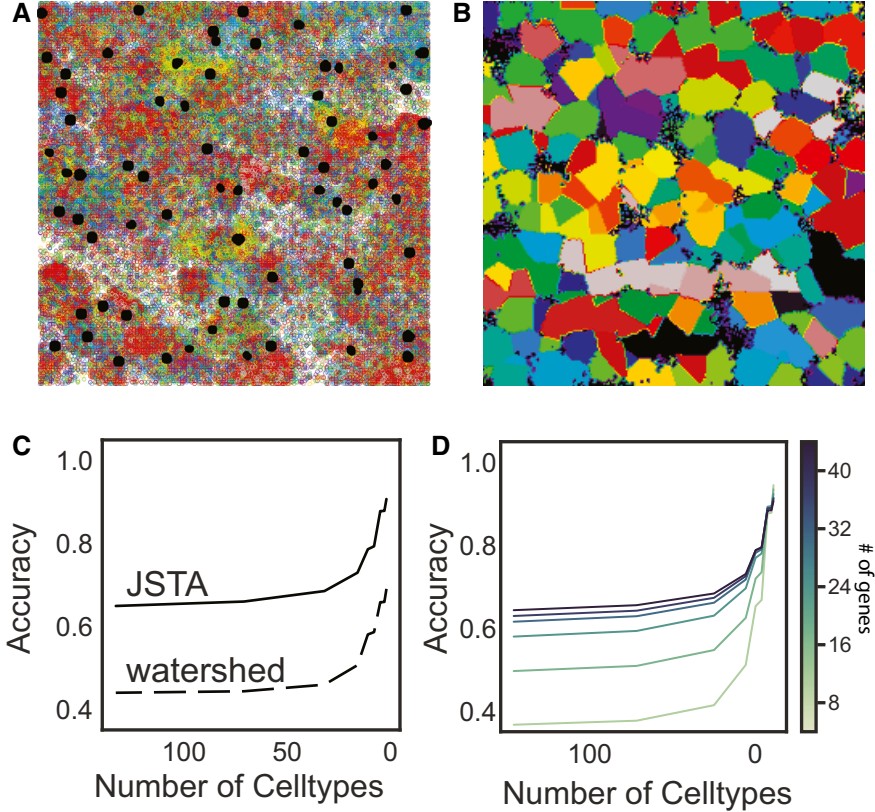

**Figure 2. Performance evaluation of JSTA using simulated data.**

A   Representative synthetic dataset of nuclei (black) and mRNAs, where each color represents a different gene.
B   Ground truth segmentation map of the cells in the representative synthetic dataset. Each color represents a different cell.
C   Average Accuracy of calling mRNA spots to cells at different cell type resolutions using 83 genes across 10 replicates. Accuracy was determined by the assignment of each mRNA molecule to the correct cell type. JSTA (solid line) is more accurate than watershed (dashed line) at assigning mRNA molecules to the correct cells (FDR < 0.05). Statistical significance was determined with a Mann–Whitney test and false discovery rate correction.
D   Accuracy (as described in (C)) of calling mRNA spots to cells when using JSTA to segment cells with a lower selection of cell type marker genes (8–44 genes tested). The color of the line gets progressively darker as the number of genes used increases.

the synthetic data showed that JSTA outperforms watershed approach, and at physiologically relevant parameters, can increase mRNA assignment accuracy by > 45%. We additionally compared JSTA to pciSeq (Qian *et al*, 2020), in the assignment of mRNA molecules to cells. We note that pciSeq is mainly designed to assign cell types to nuclei based on surrounding mRNA and therefore is not primarily focused on assigning most mRNA molecules to cells as JSTA does. Furthermore, since pciSeq is not designed to operate on 3D data, we simulated 2D data and applied both JSTA and pciSeq. We found that JSTA was more accurate at assigning mRNA molecules to cells than pciSeq (Fig EV1A). pciSeq tends to incorrectly assign many spots to background, as segmentation is not its primary goal. However, when ignoring mRNAs assigned to background in a true-positive calculation, pciSeq performs well as it primarily assigns mRNAs close to the nuclei, which is an easier task. In this case, JSTA has comparable performance (Fig EV1B).

### Time requirements of JSTA

We simulated data of different sizes and ran JSTA to determine how the run time scales with larger datasets. We simulated three replicates of data with a width and height of 100, 200, 300, 400, 500, and 1,000 μm. The run time of JSTA scales linearly with both the area and number of cells in the section (Fig EV2A and B).

### Performance evaluation using empirical spatial transcriptomics of mouse hippocampus

We next tested the performance of JSTA using empirical data and evaluated its ability to recover the known spatial distribution of coarse neuron types across the hippocampus (Fig 3). First, we subset the NCTT scRNAseq data to the shared genes we have in our MERFISH data and harmonized the MERFISH and scRNAseq datasets (Moffitt *et al*, 2018). Using the cell type annotations from the single-cell data, we trained a DNN to classify cell types. As expected, our classifier derived a cell type mapping agreeing with known spatial patterns in the hippocampus (Fig 3A). For example, CA1, CA3, and DG cells were found with high specificity to their known subregions (Fig 3B). We found that the gene expression of the segmented cells in MERFISH data highly correlated with their scRNAseq counterparts, and displayed similar correlation patterns between different cell types (Fig 3C) as seen

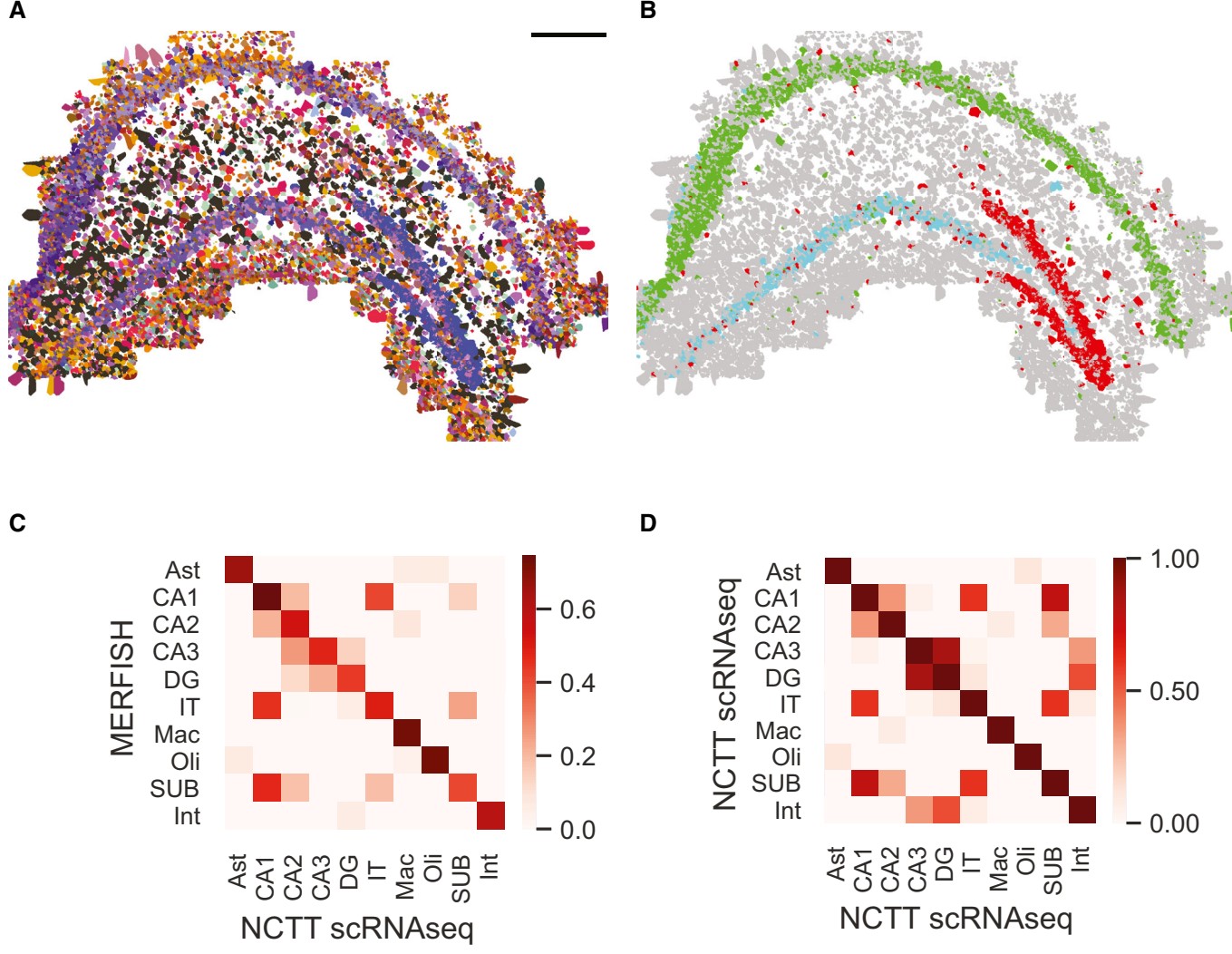

**Figure 3.   Segmentation of MERFISH data from the hippocampus using JSTA.**

A   High-resolution cell type map of 133 cell (sub)types segmented and annotated by JSTA. Colors match those defined by Neocortical Cell Type Taxonomy (NCTT). Scale bar is 500 μm.

B   JSTA-based classification of CA1 (green), CA3 (cyan), and DG (red) neurons matches their known domains.

C   Correlation of the average expression of 163 genes across major cell types between MERFISH measurements to scRNAseq data from NCTT.

D   Correlation of the average expression of the same genes as in (C) between expression of types in scRNAseq data from NCTT. The correlation structure in panel (C) closely mirrors the structure in panel (D).

in scRNAseq data (Fig 3D). These results show that our data and JSTA algorithm can recover existing knowledge on the spatial distribution of cell types and their gene expression patterns in the mouse hippocampus.

### JSTA performs high-resolution cell type mapping in the mouse hypothalamic preoptic region

We applied JSTA to a MERFISH dataset from a previously published mouse hypothalamic preoptic region with 134 genes provided (Moffitt *et al*, 2018). Using the provided scRNAseq reference dataset, we accurately mapped 87 high-resolution cell types in this region (Fig EV3A). The mapped cell types follow spatial distributions of high-resolution cell types of this region previously

annotated through clustering and marker gene annotation. We find the gene expression profiles of the cell types from the MERFISH data are highly correlated with their scRNAseq counterparts (Fig EV3B).

### JSTA performs high-resolution cell type mapping in the mouse somatosensory cortex

Next, we applied JSTA to an osmFISH dataset from the mouse somatosensory cortex with the 35 genes provided (Codeluppi *et al*, 2018). Using the NCTT reference, we mapped 142 high-resolution cell types in this region. We found that the glutamatergic neuronal populations follow known spatial organization (Fig EV4A) and that the gene expression patterns of high-resolution cell types in the

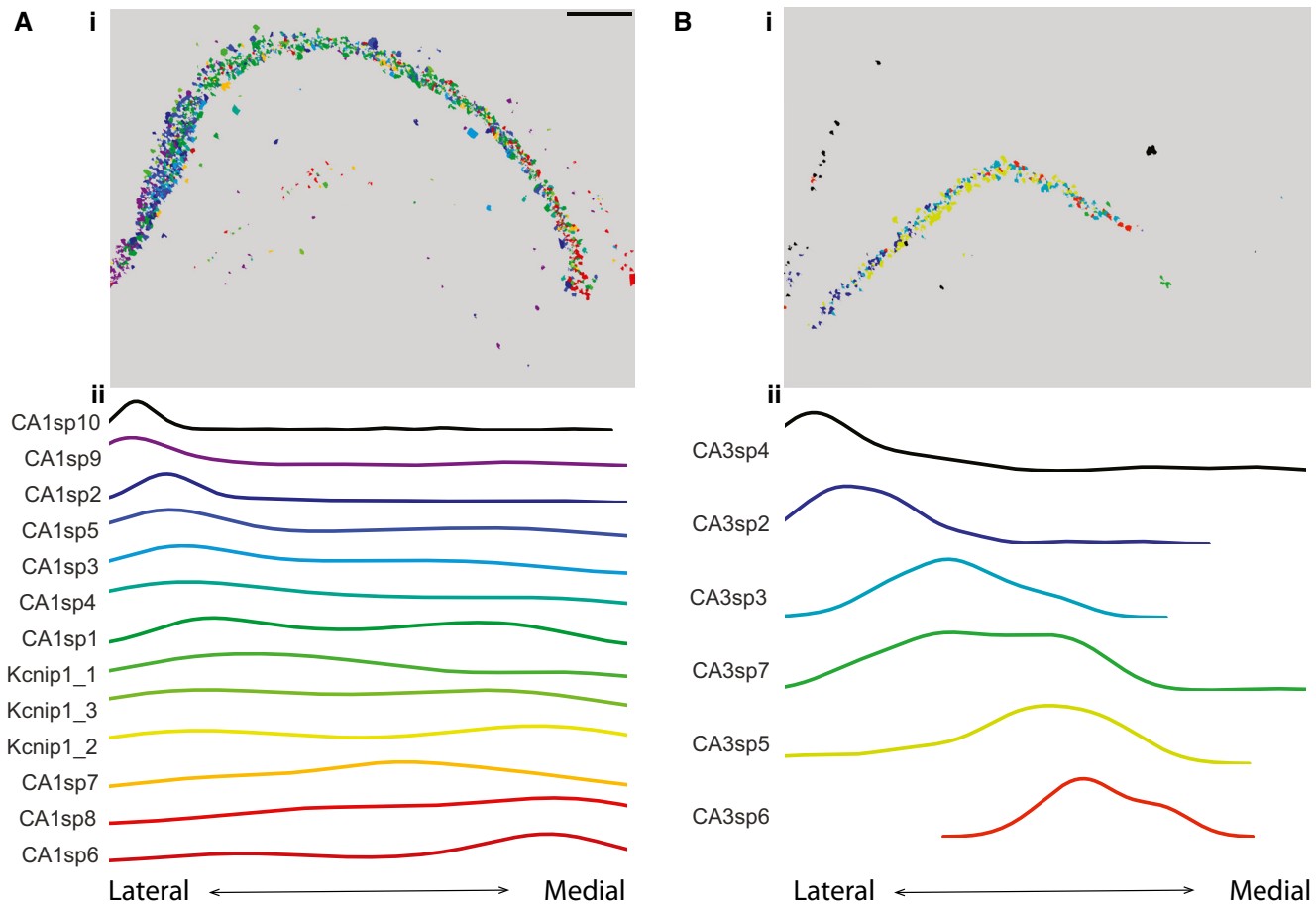

**Figure 4. Spatial distribution of neuronal subtypes in the hippocampus.**

A (i) Cell subtype map of CA1 neurons in the hippocampus as annotated by JSTA. Scale bar is 500 μm. Distribution of CA1 subtypes in the hippocampus, computed by projecting cell centers to the lateral to medial axis. CA1 neuronal subtypes show a non-uniform distribution across the whole CA1 region. (ii) Smoothed histogram highlighting the density of CA1 subtypes across the CA1 region.

B (i) Cell subtype map of CA3 neurons in the hippocampus as annotated by JSTA. Distribution of CA3 subtypes in the hippocampus, computed by projecting the cell centers to the lateral to medial axis. CA3 neuronal subtypes show a non-uniform distribution across the whole CA3 region. (ii) Smoothed histogram highlighting the density of CA3 subtypes across the CA3 region.

osmFISH data are highly correlated with their NCTT counterparts (Fig EV4B).

## Applications of JSTA for biological discovery

### JSTA identifies spatial distribution of highly granular cell (sub)types in the hippocampus

A key benefit of JSTA is its ability to jointly segment cells in images and classify them into highly granular cell (sub)types. Our analysis of mouse hippocampus MERFISH data found that these subtypes, defined only based on their gene expression patterns, have high spatial localization in the hippocampus. From lateral to medial hippocampus, the subtypes transitioned spatially from CA1sp10 to CA1sp6 (Fig 4A). Likewise, JSTA revealed a non-uniform distribution of subtypes in the CA3 region. From lateral to medial hippocampus, the subtypes transitioned from CA3sp4 to CA3sp6 (Fig 4B). This gradient of

subtypes reveals a high level of spatial organization and points to potentially differential roles for these subtypes.

### JSTA shows that spatially proximal cell subtypes are transcriptionally similar

Next, we tested whether across different cell types spatial patterns match their expression patterns by evaluating the colocalization of cell subtypes and their transcriptional similarity. Indeed, spatially proximal CA1 subtypes showed high transcriptional similarity (Figs 5A and EV5A and B). For example, cells in the subtypes CA1sp3, CA1sp1, and CA1sp6 are proximal to each other and show a high transcriptional correlation. Interestingly, this relationship was not bidirectional, and transcriptional similarity by itself is not necessarily predictive of spatial proximity. For example, subtypes CA1sp10, CA1sp7, and CA1sp4 show > 0.95 correlation but are not proximal to each other. Similar findings were seen in the CA3 region as well (Figs 5B and, EV5A and B).

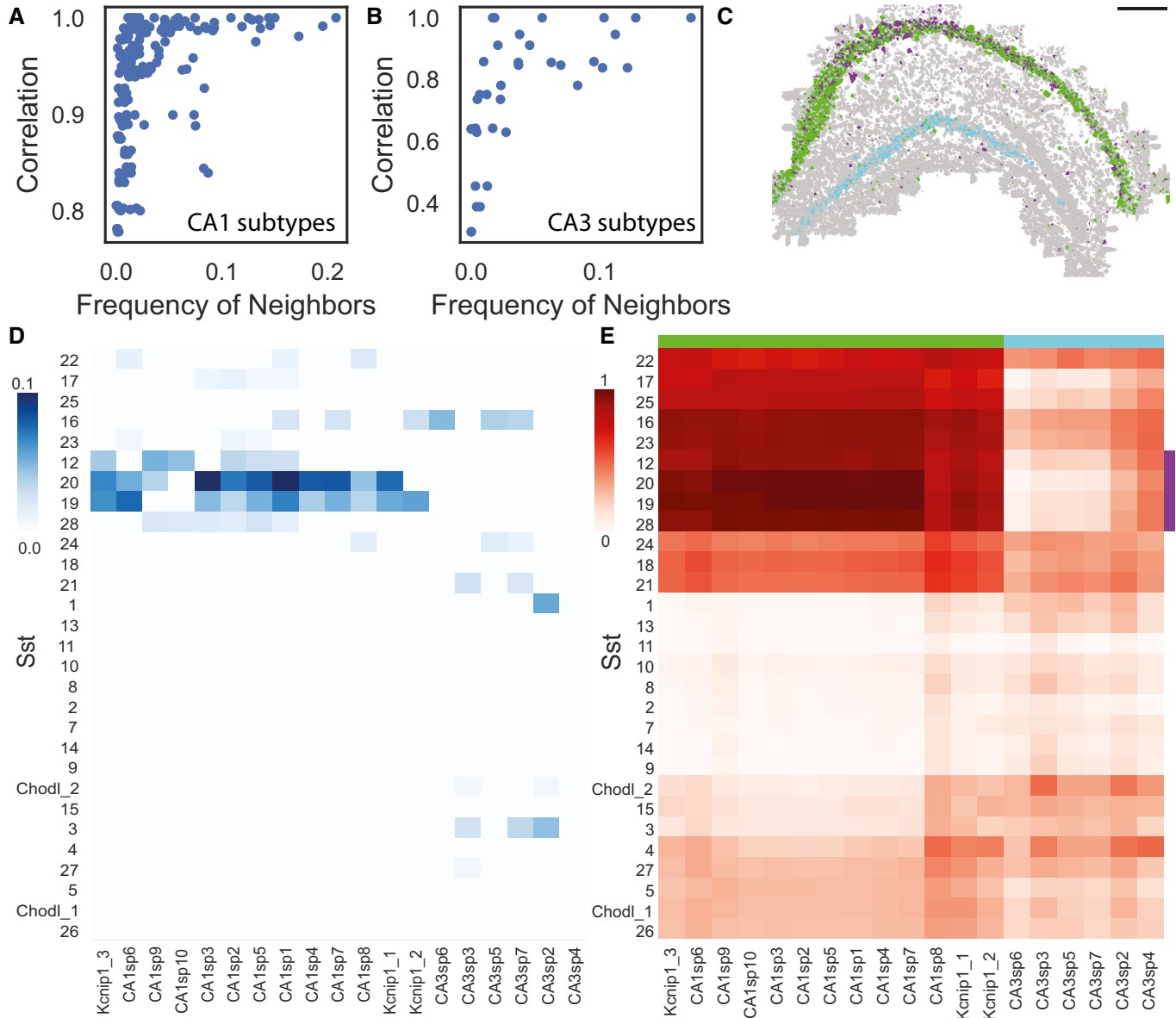

**Figure 5. Agreement between spatial proximity and gene coexpression in highly granular cell subtypes in the hippocampus.**

A, B  Relationship between the frequency of a (sub)type's neighbors and its transcriptional Pearson correlation between CA1 subtypes (A) and between CA3 subtypes (B).

C  Cell type map in the hippocampus shows specific colocalization patterns between a subset of Sst subtypes (purple) and CA1 neurons (green); these Sst subtypes do not colocalize with CA3 neurons (cyan). Scale bar is 500 μm.

D  Colocalization patterns of Sst subtypes with CA1 and CA3 subtypes. Sst subtypes that colocalize with the CA1 subtypes have high transcriptional similarity. Colocalization was defined as the percent of neighbors that are of that subtype (Materials and Methods).

E  Transcriptional correlation patterns between Sst subtypes and CA1 and CA3 neurons. Green, purple and cyan sidebars highlight the subset of Sst colocalized with CA1 (purple), CA1 (green), and CA3 (cyan).

To test whether this principle goes beyond subtypes of the same type, we compared CA1 neurons and the Sst interneurons. We found that many Sst subtypes have high specificity in their localization and are transcriptionally related to their non-Sst neighbors. Using permutation tests, we found that subtypes Sst12, Sst19, Sst20, Sst28 are significantly colocalized with these same subtypes and are specific to the CA1 region (Fig 5C and D, Materials and Methods). Analysis of their transcriptional similarity showed that these subtypes are highly correlated in their gene expression to all CA1 subtypes (Fig 5E) but not to CA3 subtypes. These results show that both within a cell type and across cell types spatial proximity indicate similarity in expression patterns.

### JSTA identifies spatial differential gene expression

Given our results on the relationship between spatial localization and gene expression patterns across cell subtypes, we next tested

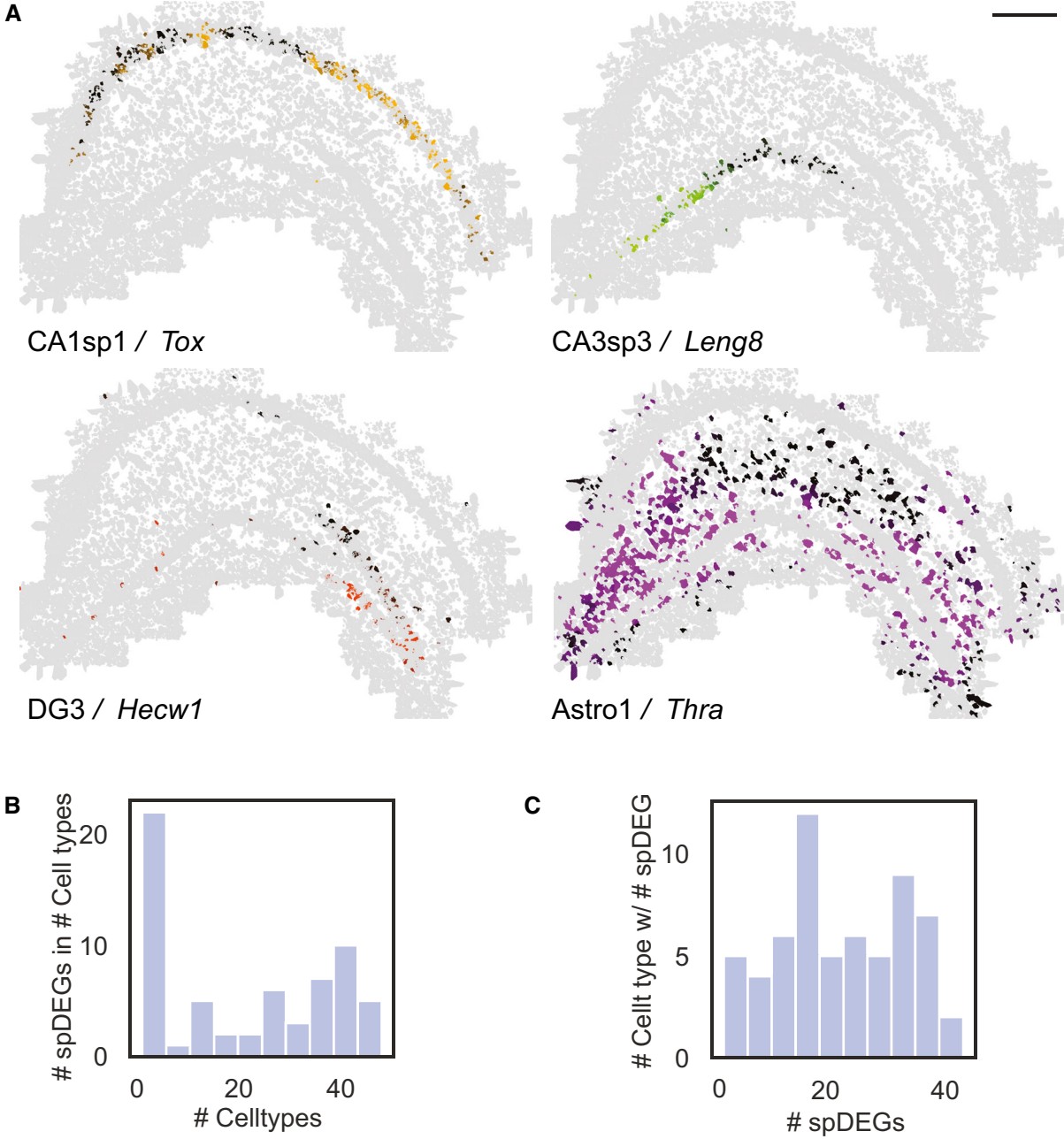

**Figure 6. Identification of spatial differential gene expression (spDEGs).**

A Normalized expression of Tox in CA1sp1, Leng8 in CA3sp3, Hecw1 in DG3, and Thra in Astro1 shows variable expression throughout the hippocampus. Scale bar is 500 μm. spDEGs were computed by comparing the true variance in gene expression between cell subtype neighborhoods to that of randomly permuted cell (sub)type neighborhoods.

B Histogram of the number of statistically significant spDEGs (Benjamini–Hochberg-corrected FDR < 0.05) in each subtype.

C Histogram of the number of subtypes that have an spDEG for each gene.

whether spDEGs within the highly granular cell subtypes can be identified. We focused our analysis on the 80 genes in our dataset that were not genes used to classify cells into cell (sub)types. We identified spDEGs by determining if the spatial expression pattern of a given gene was statistically different from a null distribution by permuting the gene expression values. Importantly, the null model was restricted to the permutation of only the cells within that subtype. As a result, our spDEG analysis specifically identifies genes whose expression within a specific subtype has a spatial distribution that is different than random. We found that within hippocampal cell subtypes, many genes were differentially expressed based on their location (Fig 6). For example, *Tox* in CA1sp1 shows higher

expression on the medial side of the hippocampus and decreases to the lateral side. *Leng8* in subtype CA3sp3 is highly expressed closer to the CA1 region and lower in the medial CA3. *Hecw1* in the DG2 subtype has varying spatial distribution in the DG region. The lower portion of the DG has clusters of higher expression, while the upper portion has lower expression. These spatial differences in gene expression are not limited to neuronal subtypes. Astrocyte subtype "Astro1" shows spatial heterogeneity in expression of *Thra*, with large patches of high expression levels and other patches of little to no expression (Fig 6A). Overall, we tested for spDEGs in 61 (sub)-types with more than 40 cells. We found that all 61 of the tested hippocampal cell subtypes have spDEGs (Figs 6B and EV6B), with more than 50% (63 of 80) of the tested genes showing non-random spatial pattern (Figs 6C and EV6C). Certain genes also show spatial patterns in many subtypes (e.g., *Thra* EV6ac), while others are more specific to a one or a few subtypes (e.g., *Farp1*, EV6ac). Identification of spDEGs highlights an interesting application of highly accurate cell type and mRNA assignment in spatial transcriptomic data.

## Discussion

Spatial transcriptomics provides the coordinates of each transcript without any information on the transcript cell of origin (Lee, 2017). Here, we present JSTA, a new method to convert raw measurements of transcripts and their coordinates into spatial single-cell expression maps. The key distinguishing aspect of our approach is its ability to leverage existing scRNAseq-based reference cell type taxonomies to simultaneously segment cells, classify cells into (sub)types, and assign mRNAs to cells. The unique integration of spatial transcriptomics with existing scRNAseq information to improve the accuracy of image segmentation and enhance the biological applications of spatial transcriptomics, distinguishes our approach from other efforts that regardless of their algorithmic ingenuity are bounded by the available information in the images themselves. As such, JSTA is not a generalist image segmentation algorithm rather a tool specifically designed to convert raw spatial transcriptomic data into single cell-level spatial expression maps. We show the benefits of using a dedicated analysis tool through the insights it provides into spatial organization of distinct (sub)types in the mouse hippocampus and the hundreds of newly discovered cell (sub)type-specific spDEGs. These insights into the molecular- and cellular-level structural architecture of the hippocampus demonstrate the types of biological insights provided by highly accurate spatial transcriptomics.

The promise of single cell and spatial biology lends itself to intense focus on technological and computational development and large-scale data collection efforts. We anticipate that JSTA will benefit these efforts while at the same time benefit from them. On the technology side, we have demonstrated the performance of JSTA for two variants of spatial transcriptomics, MERFISH and osmFISH. However, the algorithm is extendable and could be applied to other spatial transcriptomic approaches that are based on in situ sequencing (Lee *et al*, 2014; Lee *et al*, 2015; Turczyk *et al*, 2020), subcellular spatial barcoding (Ståhl *et al*, 2016; Salmén *et al*, 2018), and potentially any other spatial "omics" platforms (Gerdes *et al*, 2013; Lin *et al*, 2015; Goltsev *et al*, 2018; Keren *et al*, 2018; Lin *et al*, 2018;

Lundberg & Borner, 2019). Additionally, cell segmentation results from JSTA can be used as input for other tools such as GIOTTO (Dries *et al*, 2021) and TANGRAM (preprint: Biancalani *et al*, 2020) to facilitate single cell and spatial transcriptomic data analysis. The benefits of JSTA are evident even with a small number of measured genes. This indicates that it is applicable to a broad range of platforms across all multiplexing capabilities. JSTA is limited by its ability to harmonize technical differences between spatial transcriptomic data modalities and the scRNAseq reference. Harmonization between datasets is an active area of research, and JSTA will benefit from these advances (preprint: Lopez *et al*, 2019; Stuart *et al*, 2019; Welch *et al*, 2019; Abdelaal *et al*, 2020; Tran *et al*, 2020). JSTA relies on initial seed identification (nuclei or cell centers), and incorrect identification can lead to split or merged cells. JSTA currently does not split or merge cells, but this postprocessing step could be added to further improve segmentation (Chaudhuri & Agrawal, 2010; Surut & Phukpattaranont, 2010; Correa-Tome & Sanchez-Yanez, 2015; Gamarra *et al*, 2019). On the data side, as JSTA leverages external reference data, it will naturally increase in its performance as both the quality and quantity of reference cell type taxonomies improve (HuBMAP Consortium, 2019). We see JSTA as a dynamic analysis tool that could be reapplied multiple times to the same dataset each time external reference data is updated to always provide highest accuracy segmentation, cell (sub)type classification, spDEG identification.

Due to the nascent status of spatial transcriptomics, there are many fundamental questions related to the interplay between cell (sub)types and other information gleaned from dissociative technologies and tissue and organ architecture (Trapnell, 2015; Mukamel & Ngai, 2019). Our results show that strong codependency between spatial position and transcriptional state of a cell in the hippocampus, these results mirror findings from other organs (Halpern *et al*, 2017; Moor *et al*, 2018; Egozi *et al*, 2020). This codependency supports the usefulness of the reference taxonomies that were developed without the use of spatial information. Agreements between cell type taxonomies developed solely based on scRNAseq and other measurement modalities, i.e., spatial position, corroborate the relevance of the taxonomical definitions created for mouse brain (Yuste *et al*, 2020). At the same time, the spatial measurements demonstrate the limitation of scRNAseq. We discovered many spatial expression patterns within most cell (sub)types that prior to these spatial measurements would have been considered biological heterogeneity or even noise but in fact they represent key structural features of brain organization. High accuracy mapping at the molecular and cellular level will allow us to bridge cell biology with organ anatomy and physiology pointing toward a highly promising future for spatial biology.

## Materials and Methods

### Tissue preparation

All experiments were performed in accordance with the United States National Institutes of Health Guide for the Care and Use of Laboratory Animals and were approved by the University of California at Los Angeles Chancellor's Animal Research Committee. B6 mouse was euthanized using carbon dioxide with cervical dislocation. Its brain was harvested and flash-frozen in Optimal Cutting

Temperature Compound (OCT) using liquid nitrogen. 15 μm sections were prepared and placed on pretreated coverslips.

## Coverslip functionalization

Coverslips were functionalized to improve tissue adhesion and promote gel attachment (Moffitt & Zhuang, 2016). Briefly, 40 mm No.1 coverslips were cleaned with a 50:50 mixture of concentrated 37% hydrochloric acid and methanol under sonication for 30 min. Coverslips were silanized to improve gel adhesion with 0.1% triethylamine and 0.2% allyltrichlorosiloxane in chloroform under sonication for 30 min then rinsed once with chloroform then twice with ethanol. Silanization was cured at 70°C for 1 h. An additional coating of 2% aminopropyltriethoxysilane to improve tissue adhesion was applied in acetone under sonication for 2 min then washed twice with water and once with ethanol. Coverslips were dried at 70°C for 1 h then stored in a desiccator for less than 1 month.

## Probe design and synthesis

A total of 18 readout probes were used to encode the identity of each gene. Each gene was assigned four of the possible 18 probes such that each combination was a minimum hamming distance of 4 away from any other gene. This provides classification that is robust up to 2-bit errors. 80–120 encoder probes were designed for each target gene. Encoder probes contained a 30 bp region complementary to the transcript of interest with a melting point of 65°C and less than 17 bp homology to off-target transcripts including highly expressed ncRNA and rRNA. Probes also contained three of four readout sequences assigned to each gene. Sequences are available in supplementary material. Probes were designed using modified MATLAB code developed by the Zhuang Lab (Moffitt & Zhuang, 2016). Probes were ordered from custom arrays as a single strand pool. A T7 promoter was primed into each sequence with a limited cycle qPCR to allow amplification through *in vitro* transcription and reverse transcription (Moffitt & Zhuang, 2016).

## Hybridization

Hybridization was performed using a modified MERFISH protocol (Moffitt & Zhuang, 2016). Briefly, tissue sections were fixed in 4% PFA in 1xPBS for 15 min and washed three times with 1×PBS for 5 min each. Tissue was permeabilized with 1% Triton X-100 in 1×PBS for 30 min and washed three times with 1×PBS. Tissue was incubated in 30% formamide in 2×TBS at 37°C for 10 min. Encoding probes were hybridized at 5 nM per probe in 30% formamide 10% dextran sulfate 1 mg/ml tRNA 1 μM poly-T acridite anchor probed and 1% murine RNAse inhibitor in 2xTBS. A 30 μl drop of this encoding hybridization solution was placed directly on the coverslip, and a piece of parafilm was placed on the coverslip to prevent evaporation. Probes were hybridized for 30–40 h at 37°C in a humidity chamber. Tissue was washed twice with 30% formamide in 2×TBS for 30 min each at 45°C. Tissue was washed three times with 2×TBS. Tissue was embedded in a 4% polyacrylamide hydrogel with 0.5 μl/ml TEMED 5 μl 10% APS and 200 nm blue beads for 2 h. Tissue was cleared with 1% SDS, 0.5% Triton x-100, 1 mM EDTA, 0.8 M guanidine HCl 1% proteinase K in 2×TBS for 48 h at 37°C replacing clearing solution every 24 h. Sample was washed with 2×TBS and mounted for

imaging. Readout hybridization was automated using a custom fluidics system. Sample was rinsed with 2×TBS and buffer exchanged into 10% dextran sulfate in 2×TBS for hybridization. Hybridization was performed in 10% dextran sulfate in 2×TBS with a probe concentration of 3 nM per probe. Sample was washed with 10% dextran sulfate then 2×TBS. Sample chamber was filled with a 2 mM pca 0.1& rPCO 2 mM VRC 2 mM Trolox in 2×TBS Imaging Buffer. Sample was imaged at 63× using a custom epifluorescent microscope. After imaging, fluorophores were stripped using 50 mM TCEP in 2×TBS and the next round of readout probes was hybridized.

## Image analysis

Image analysis was performed using custom python code (Wollman lab). To register multiple rounds of imaging together with subpixel resolution, fiduciary markers were found and a rigid body transformation was performed. Images were preprocessed using hot pixel correction, background subtraction, chromatic aberration correction, and deconvolution. An 18-bit vector was generated for each pixel where each bit represented a different round and fluorophore. Each bit was normalized so that background approached 0 and spots approached 1. An L2 normalization was applied to the vector, and the Euclidean distance was calculated to the 18-bit gene barcode vectors. Pixels were classified if their Euclidean distance was less than a 2-bit error away from the nearest gene barcode. Individual pixels that were physically connected were merged into a spot. Dim spots and spots that contained 1 pixel were removed.

## Nuclei segmentation

Nuclei were stained using dapi and imaged after MERFISH acquisition. Each 2D image was segmented using cellpose with a flow threshold of 1 and a cell probability threshold of 0 (preprint: Stringer *et al*, 2020). 2D masks of at least 10 μm$^2$ area were merged if there was at least 30 percent overlap between frames. 3D masks that were present in < 5 *z* frames (2 μm) were removed.

## Simulation

### scRNAseq reference preparation

The NCTT was subset to the cells found in the hippocampus and to the genes from our MERFISH data. Expression levels of simulated genes were taken from scRNAseq reference and were harmonized to qualitatively match the variance observed in measured in MERFISH data. These were then rounded to create a scaled count matrix. For each of the 133 hippocampal cell types from the NCTT, we computed a mean vector and covariance matrix of gene expression. We additionally computed the cell type proportions in the single-cell data for later use in cell type assignment.

### Creating the cell map

Initially, the cell centers were placed in a 200 × 200 × 30 μm grid, equidistant from one another, with an average distance between cell centers of 4 μm. The cell centers were then moved around in each direction (*x*, *y*, *z*) based on a Gaussian function with mean 0 and standard deviation 0.6. Pixels were then assigned to their closest center with a minimum distance of 5 μm and maximum distance of 7 μm. Cells with less than 30 pixels were removed due to small

unrealistic sizes. To create more realistic and non-round cells, we merged neighboring, touching cells twice. Each cell was assigned a (sub)type uniformly across all 133 types in our dataset. Nuclei were randomly placed within each cell with 20 pixels. Nuclei pixels placed on the border were removed. We simulated 10 independent replicates in each simulation study.

### Generating cell transcriptional profiles and placing spots
Each cell's gene expression profile was drawn from a multivariate Gaussian using the mean vector, and covariance matrix computed from the scRNAseq reference. This vector and matrix are cell type specific, and each cell's gene expression profile is sampled from these cell type-specific distributions. The mRNA spots were then placed inside of each cell, slightly centered around the nucleus, but mostly uniform throughout.

### Simulated data on limited genes
To perform feature selection and extract a limited number of important genes (4, 12, 20, 28, 36, 44), we used a random forest classifier with 100 trees to predict cell types in the reference dataset. The top $n$ important features for classifying cell types were used. Other simulation parameters were the same as above.

### K-nearest neighbor-based density estimation method

We used a K-nearest neighbor approach to estimate density for many genes at each point (Wasserman, 2006). The volume required to reach the 5[th] spot was computed and used to compute the density estimation (equation 1). Where $r$ is the radius to the 5[th] closest spot of that gene, we repeated this process for all genes.

$$\text{density} = \frac{5}{\frac{4}{3}\pi r^3} \tag{1}$$

### JSTA overview

Expectation maximization can be used to jointly classify the identity of an observation of interest, while learning the parameters that describe the class distributions. In EM, the object classes are initialized with a best guess. The parameters of the classifying function are learned from this distribution of initialized classes (M-step). The objects are reclassified according to the updated function parameters (E-step). These steps are repeated until the function parameters converge. JSTA is designed with an EM approach for reclassifying border pixels in the 3-dimensional grid of pixels based on their estimated transcriptional densities. First, we initialize the spatial map with watershed, in Euclidean space with a maximum radius. Next, we classify cell types of the segmented cells based on the computed count matrix. We then randomly sample a fraction of the pixels' gene expression vectors, and train a pixel classifier (M-step). The pixel classifier is used to reclassify the cell identity of pixels that are at the border between different cell types, or between a cell and empty space (E-step).

### Cell type classification

#### Data preparation
To match the distributions of both scRNAseq and MERFISH, we centered and scaled each cell across all genes. We then subsequently centered and scaled each gene across all cells. We note that other harmonization approaches could be applied here.

### Cell type classifier
We parameterized the cell type classifier as a neural network, with three intermediate layers with three times the number of input genes as nodes. We used a tanh activation function with L1 regularization (1e-4) allowing for the influence of negative numbers in the scaled values and parameter space sparsity (preprint: Bach *et al*, 2011). Batch normalization was used on each layer (preprint: Ioffe & Szegedy, 2015), and a softmax activation was used for the output layer (Goodfellow *et al*, 2016) (Table EV1).

### Training the classifier
The network parameters were initialized with Xavier initialization (Glorot & Bengio, 2010). The neural network was trained with two steps with learning rates of 5e-3 and 5e-4 for 20 epochs each, with batch size of 64, and the Adam optimizer was used (preprint: Kingma & Ba, 2014). A 75/25 train validation split was used to tune the L1 regularization parameter and reduce overfitting. We used 75/25 to increase the representation of lower frequency cell classes. Cross-entropy loss was used to penalize the model and update parameters accordingly (Fig EV7A and B).

### Pixel classification

### Pixel classifier
We parameterized the pixel classifier as a neural network with three intermediate layers. Each layer was twice the size of the last to increase the modeling power of this network and indirectly model the other genes not in the MERFISH dataset. Each layer used the tanh activation function and used an l2 regularizer (1e-3). Each layer was centered and scaled with batch normalization, and the output activation was an l2 regularized softmax function (Table EV2).

### Training the classifier
Each time cell types are reclassified, a new network was reinitialized with Xavier initialization. The network was initially trained with learning rates or 1e-3 and 1e-4 for 25 epochs. After the first round of classifying and flipping the assignment of pixels, the network was retrained on a new sample of pixels starting from the previous parameter values. This was then trained with a learning rate of 1e-4 for 15 epochs. All training was performed with the Adam optimizer and a batch size of 64. We used an 80/20 train validation split to help monitor any overfitting that might be occurring, and adjust the hyperparameter selection accordingly. We used cross-entropy loss (Fig EV7C and D).

### Identifying border pixels
Border pixels are defined as pixels that are between two cells of different types, or between a cell and empty space. To enhance the smoothness of cells' borders, we require a border pixel to have 5 of its surroundings be from a different cell, and 2 of its surroundings be from the same cell.

### Classifying pixels
The trained classifier was then used to estimate the cell type class of border pixels. The pixel classifier outputs a probability vector for each cell type, and the probabilities are scaled by a distance metric

based on the distance to the cells' nuclei that it could flip to. Probabilities less than 0.05 are set to 0. The classification is sampled from that probability vector subset to cell types of its neighbors, and renormalized to 1. If the subset probability vector only contains 0, the pixel identity is set to background. To balance the exploration and exploitation of pixel classification map, we anneal the probability of selecting a non-maximum probability cell type by multiplying the maximum probability by (1 + number of iterations run*0.05). If this is selected as 0, complete stochasticity presides, and if it is large, the maximum probability will be selected.

## JSTA formalization

### Definitions and background

The gene expression level of $n_c$ cells and $n_p$ pixels is described by the matrices $E_c$ (cells) and $E_p$ (pixels) which are $n_c \times m$ and $n_p \times m$ matrices, respectively, where $m$ is the number of genes. Likewise, cell type probability distributions of all cells or pixels can be described by matrices. These distributions for cells and pixels are $P_c$ and $P_p$, respectively, represented as $n_c \times k$ and $n_p \times k$ matrices, where $k$ is the number of cell types. We aim to learn $\theta$ and $\phi$, such that $f_\theta$ and $g_\phi$, accurately map from $E_c$ to $P_c$ and $E_p$ to $P_p$. We used the cross-entropy loss function for penalizing our models.

### Cell type classification

First, we learn the parameters of $f_\theta$ by:

$$\theta = \arg\min_\theta \left[ \text{Loss}\left( f_\theta\left(E_{ref}\right), T_{ref} \right) \right]$$

where $E_{ref}$ is an $n_{ref} \times m$ gene expression matrix representing the harmonized NCTT data and $T_{ref}$ is an $n_{ref}$ vector of cell type labels provided by NCTT. We then use the newly learned mapping to infer the cell type probability distributions in the initialized dataset $E_c$ with:

$$P_c = f_\theta(E_c).$$

We classify each cell as the highest classification probability for that cell:

$$T_c = \arg\max_k(P_c)$$

where $T_c$ are the predicted cell types for each of the cells in the matrix $E_c$.

### Joint pixel and parameter updates

We initialize the labels $T_p$ for all pixels based on the current segmentation map that assigns pixels to cells. We then learn the parameters of the mapping function $g_\phi$ (maximization). Learning is performed by updating the parameters of the mapping function $g_\phi$ with:

$$\phi = \arg\max_\phi \left[ \text{Loss}\left( g_\phi\left(E_p\right), T_p \right) \right].$$

The updated mapping function is then used to infer the probability of observing a type $T_p$ given expression $E_p$ in all pixels:

$$P_p = P\left(T_p | E_p\right) = g\left(E_p\right).$$

The next step is to update $P_p$ based on spatial proximity to cells of each type. Using the notation $q$ for the vector of probabilities of a single pixel ($q = P_{pj} = [q_0 \dots, q_i, \dots q_k]$), we next update the elements in the vector $q$ based on neighborhood information. We scaled the values of $q_i$ based on its distance from the nuclei and its neighbors. $q'$ is intermediate in the calculation that does not represent true probabilities.

$$q'_i = \begin{cases} 10 & r < d \\ \dfrac{q_i * d * 5}{2(r-d)} & r \ge d \end{cases}$$

where $r$ is the distance from the nucleus of the closest cell of cell type $i$, $d$ is the distance threshold for which a pixel should automatically be assigned to that nucleus. The values 10 and 5 were determined empirically to modify the sharpness of probability decline based on distance. 10 was chosen to be much bigger than probabilities produced by $g_\phi$, and 5 was chosen to allow the probability to decay to half over $5d$.

We then only kept probabilities for cell types of neighboring cells:

$$q'_i = \begin{cases} 0 & \text{if } i \text{ is neighbor} \\ q'_i & \text{otherwise} \end{cases}.$$

We then used the intermediate $q'$ to recalculate the pixel type probabilities:

$$q_i = \frac{q'_i}{\sum_{i=o}^{i=k} q'_i}$$

The updated values per cell ($q_j$) are then used to update the probability matrix $P_p$. The type per pixel ($T_p$). The assignment of pixel to cells is then stochastically assigned according to the inferred probability $P_p$ per pixel basis.

$$T_{pj} \sim \text{multinomial}\left(P_{pj}\right).$$

We then repeat updating $g_\phi$ and $T_p$ until convergence.

## Segmentation

### Density estimation

The 3-dimensional space was broken into a grid of pixels with the edge of each pixel 2 μm in length (1 μm in simulation). The density was estimated at the center of each pixel, for each gene. The volume required to reach five mRNA molecules was used as the denominator of the density estimation.

### Segmentation with JSTA

The cell assignment map was initialized with watershed on the distance transform with a maximum distance from the nucleus of 2 μm. The cells were only classified once. The pixel classifier was trained six times (5 in simulation) on 10% of the pixels excluding pixels without assignment. After each training step, we reassigned pixels for 10 iterations (5 in simulation). The lowest probability kept in the predicted pixel assignment vector was 0.05 (0.01 in simulation).

### Segmentation with watershed

The overall gene density was the sum of each gene in a given pixel. To smooth the range of the density, we $\log_2$ transformed

the density values. Log-transformed density values less than 1 were masked. The segmentation used the nuclei locations as seeds and watershed from the skimage python package, with *compactness* of 10. Using compactness of 10 was the highest performing value for watershed. A watershed line was used to separate cells from one another.

## Evaluation of segmentation in simulated data

mRNA spot call accuracy was evaluated at different taxonomic levels. For a given cell, the accuracy was defined as the number of mRNA spots correctly assigned to that cell divided by the total number of mRNA spots assigned to that cell. To match the algorithm's ability to segment based on cell type information, RNAs that were assigned to a neighboring cell of the same (sub)type were also considered correct assignment. The overall segmentation accuracy was the mean accuracy across all cells in a given sample. To evaluate accuracy at different levels, we utilized the NCTT dendrogram. We used dendrogram heights at 0 through 0.8 with a step size of 0.05 (133, 71, 32, 16, 11, 8, 5, 4, 3, 2 cell types).

## Correlation of segmented MERFISH with scRNAseq

The NCTT scRNAseq data were subset to the genes from our MERFISH data. Cells in the segmented MERFISH dataset were assigned to canonical hippocampus cell types (Astrocyte, CA1 pyramidal neuron, CA2 Pyramidal neuron, CA3 Pyramidal, Dentate Gyrus, Inferior temporal cortex, Macrophage, Oligodendrocyte, Subiculum, Interneuron) based on their high-resolution cell type classification. In each cell type, the average expression in each gene was calculated. Only genes were kept that had an average expression of at least five counts in one of the cell types. Values were centered and scaled across all cell types. The Pearson correlation was computed for each gene for the matching cell types between scRNAseq and MERFISH.

## Distribution of high-resolution cell types in the hippocampus

CA1 and CA3 subtypes were projected onto the lateral medial axis. The smoothed density across this dimension was plotted for each of the subtypes.

## Colocalization of high-resolution cell types

Significant colocalization of subtypes was determined through a permutation test. First, the 20 nearest cell types around each cell were determined. We counted the number of cells from each type that surround each cell type and computed the fraction of neighbors coming from each subtype. This created a matrix with the fraction of colocalizations per cell between each cell type combination. We then permuted the labels of the cell types 1,000 times and recomputed this interaction matrix to create a null distribution. For each cell type colocalization, we determined the percentage of colocalizations in the null distribution that is higher than the true colocalization number to create a *P*-value for each colocalization. We corrected for multiple testing with the Benjamini–Hochberg procedure and determined significance using FDR < 0.05.

## Identification of spatial differential gene expression

spDEGs were calculated in cell types with more than 40 cells. Within each cell type, we computed a local expression of each gene for each cell. The local expression was the mean expression of a gene in the cell and its nine nearest neighbors. We then built a null distribution by permuting gene expression values within the cell type, and repeating the local expression process for 100 permutations. Determining if a gene was spatially differentially expressed, we compared the variance of the null distribution within a cell type with the variance of the true distribution of local expression to get a *P*-value. We corrected for multiple testing with Benjamini–Hochberg procedure and determined significance using FDR < 0.05.

## Python packages used

python (3.8.3), numpy (1.18.5), pandas (1.0.5), matplotlib (3.2.2), scipy (1.5.0), scikit-learn (0.23.1), scikit-image (0.16.2), tensorflow (2.2.0). seaborn (0.10.1).

# Data availability

Source code: GitHub (https://github.com/wollmanlab/JSTA; https://github.com/wollmanlab/PySpots).

Raw images: Figshare (https://doi.org/10.6084/m9.figshare.1453 1553).

**Expanded View** for this article is available online.

## Acknowledgement
The work was funded by NIH grant R01NS117148 and T32CA201160.

## Author contributions

RL, XY, and RW developed the algorithm that was implemented by RL. DA and RF designed MERFISH probeset. ZH and RF performed MERFISH measurements and initial image analysis. GZ and FG-P performed brain sample preparation.

## Conflict of interest
The authors declare that they have no conflict of interest.

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
