## [Review Process File · Molecular Systems Biology]

Joint cell segmentation and cell type annotation for spatial transcriptomics

Russell Littman, Zachery Hemminger, Robert Foreman, Douglas Arneson, Guanglin Zhang, Fernando Gomez-Pinilla, Xia Yang, and Roy Wollman

DOI: [10.15252/msb.202010108](https://doi.org/10.15252/msb.202010108)

Corresponding author(s): Roy Wollman (rwollman@ucla.edu) , Xia Yang (xyang123@ucla.edu)

Review Timeline:

Submission Date:	6th Nov 20
Editorial Decision:	26th Nov 20
Revision Received:	8th Mar 21
Editorial Decision:	13th Apr 21
Revision Received:	4th May 21
Accepted:	6th May 21

Editor: Maria Polychronidou

Transaction Report:

Thank you again for submitting your work to Molecular Systems Biology. We have now heard back from the three referees who agreed to evaluate your study. Overall, the reviewers acknowledge that the proposed method seems potentially interesting. However, they raise a series of concerns, which we would ask you to address in a major revision.

Without repeating all the points listed below, some of the more fundamental issues are the following:

- The reviewers refer to the need to include proper benchmarking and direct comparisons to other methods. Such analyses are very important in order to better support the methodological advance and the improvements/advantages compared to existing approaches.
- Reviewer #2 points out that applications to further datasets would increase the confidence in the performance of the method.
- The reviewers also raise several technical concerns, which need to be addressed.

Please let me know in case you would like to discuss in further detail any of the issues raised. All issues raised by the referees would need to be satisfactorily addressed. As you might already know, our editorial policy allows in principle a single round of major revision so it is essential to provide responses to the reviewers' comments that are as complete as possible.

On a more editorial level, we would ask you to address the following.

REFeree REPORTS

Reviewer #1:

The authors present a method to segment in situ transcriptomic data to cells by using prior knowledge from scRNA sequencing clusters of cells from the same source of tissue. The approach is conceptually very similar to one of the cited methods, pciSeq, but the authors have provided an alternative method to achieve the same objectives. The authors need to improve the description and distinction of prior work, as well as, bench-mark their method to prior relevant methods. In the introduction they write "both pciSeq and SSAM are limited to cell type information and do not create a segmentation map for the assignment of genes that are not cell type markers" This sentence needs clarification. Both JSTA and pciSeq use cell type definition from single cell RNAseq to segment cells, and use all in situ data to do that.

It's a bit unclear what the distinction of marker genes and non-marker genes is in the results section. I assume that the marker genes are the ones that have implications in setting the cell boundaries in their model, and the non-marker genes are the ones that are "assigned" to a cell based on the segmentation map. The question is: how do you decide that a gene is a marker gene or not? Most genes are differentially expressed between cells to some extent.

An important pre-requirement for this method is the necessity of harmonizing the scRNAseq data and the spatial dataset. It would be interesting to see some simulations and discussion of potential pit-falls in this harmonization. How does variation in data density in the scRNA seq data set, and the in situ data set affect reproducibility of this harmonization?

Since the method is based on an initial watershed segmentation with a decent quality, it would be interesting to know what happens with cells where this segmentation goes wrong. Is the algorithm able to split cells incorrectly merged by watershed, or conversely, join incorrectly split cells?

When evaluating the performance of the JSTA method, it would be very useful to include a comparison to how other published segmentation methods performs. There are several out there now, in addition to pciSeq and SSAM, there is e.g. spage2vec

(<https://febs.onlinelibrary.wiley.com/doi/10.1111/febs.15572>). A minimum requirement would be to benchmark with pciSeq since they are conceptually very similar.

Figure 4: What's the x-axis?.

In the discussion, the authors mention that JSTA could be applied to other spatial technologies, and mention spatial barcoding approaches, referring to ST/Visium. I do not see how this could be done.

Reviewer #2:

The submitted manuscript MSB-2020-10108 in Molecular Systems Biology entitled 'Joint cell segmentation and cell type annotation for spatial transcriptomics' by Littmann and colleagues describes and applies their Joint cell Segmentation and cell Type Annotation (JSTA) analysis workflow to MERFISH spatial transcriptomics data of the mouse hippocampus. It combines an initial watershed segmentation algorithm with mRNA-based pixel classifiers to assign cell types for spatial transcriptomics, whereas JSTA derives pixel probabilities from deep neural network cell and pixel classifiers. By spatial differential gene expression segmented mouse hippocampus cells are further stratified using non-cell type markers. Although the JSTA method seems to be conceptually useful, any comparison is missing.

Major concerns:

1. The manuscript is based on only one data set (MERFISH). It would be recommended if Littmann et al. evaluate JSTA also one other data set to proof the concept. Baysor (<https://www.biorxiv.org/content/10.1101/2020.10.05.326777v1>) and SSAM use e.g. osmFISH and MERFISH. Please introduce Baysor in the intro as well. Different alternative other approaches are also mentioned in the discussion as applications (ISS and spatial barcoding).
2. The intro about other analysis pipelines is at least misleading. Their introduction: "both pciSeq and SSAM are limited to cell type information" is not true as modelling non-marker genes is possible, but not shown. ", and do not create a segmentation map for the assignment of genes that are not cell type markers". This is unfair to pciSeq, as this assigns cell-types to cells (based on segmentation). Please carefully rewrite the whole paragraph similar to 'pciSeq assigns cell types to prior segmented regions based on proximity to mRNA of marker genes, circumventing the need for pixel level segmentation. Similarly, SSAM creates cell type maps based on RNA distributions without needing to know cellular boundaries. However, both pciSeq and SSAM do not create a segmentation map. Therefore, while both pciSeq and SSAM leverage cell type catalogs to provide insights into the spatial distribution of different cell types they do not produce a cell segmentation map and are only demonstrated on cell (sub)type label information (cell type map).' Again, the description of not being able to consider non-marker genes is inaccurate. All this might change the impact/uniqueness of the manuscript (including Baysor, Petukhov et al.).
3. Consequently, the spDEG of JSTA in Fig. 6 (SFig. 3) with 50% of tested genes showing a non-random distribution is probably not an exclusive feature of JSTA. It is unclear how other methods mentioned in the introduction or application would perform here.

Minor concern:

1. Please add page numbers.
2. Please revise axis labels Fig. 3d? Maybe Fig. 3 subfigure without any JSTA segmentation in correlation with NCTT would be useful to see the advances.
3. Overall labels, abbreviations and headers in figure legends should be more detailed.

Overall, the manuscript leaves the reader with the question: how significant are these technical advances in comparison?

Reviewer #3:

This paper presents an optimization approach for segmentation and annotation of single cells from spatial transcriptomics datasets. Unlike conventional approach using watershed, the authors propose a pixel based density and optimization approach for classification of transcript data to associate cell subtypes based on the known expression of individual cells.

The JSTA approach is interesting, however, the following can be addressed to improve the paper:

- 1) Nowadays the gold standard for cell segmentation and annotation is to also perform protein staining on the same samples after the MERFISH experiments. The cell boundaries using FISH labels can be misleading as the density of RNA molecules is highly variable. Authors should validate their segmentation results by comparing the data to a cell mask that is generated from a surface protein staining.
- 2) The number of transcripts per pixel would change based on the biological significance of each gene for unique cell types. Would this create bias in the annotation and segmentation of the JSTA

approach?

3) The pixel distributions of each pixels would be beyond 3x3 pixels due to diffraction limit and overlapping RNA molecules would be also making the assignment of specific RNA type complicated. Could authors change the pixel window from 3x3,5x5, and 7x7 etc. to fine tune their results that can potentially reduce the effect of diffraction limit.

4) How would this approach compare to a deep learning framework (<https://www.nature.com/articles/s41592-019-0403-1>) that can be applied to either pixel data or cells directly? Would a different model e.g., Keras improve these segmentation results?

5) Which dyes were used for these MERFISH experiments. Is it only Alexa 647 or multiple colors? If so, how would the authors address the pixel level shifts of transcripts due to the optical aberrations?

6) How would this JSTA scale up with data size? If the entire mouse brain were imaged, would it be computationally efficient to run these classifiers in a reasonable time. In other words, how long does it take and at which computation speed?

7) Finally, while they are not the same, how would this approach complement GIOTTO (<https://www.biorxiv.org/content/10.1101/701680v1.full>) and TANGRAM (<https://www.biorxiv.org/content/10.1101/2020.08.29.272831v3>)? What is the advantage of the JSTA compared to the recent emerging literature including these two recent platforms?

The cell segmentation and annotation issues are primary concerns of cell atlas studies that can impact the understanding of basic mouse biology and human health. These solutions will be highly important to enable reproducible and efficient data analysis pipelines for spatial transcriptomics technologies.

Reviewer #1:

1.1. The authors present a new method to segment in situ transcriptomic data to cells by using prior knowledge from scRNA sequencing clusters of cells from the same source of tissue. The approach is conceptually very similar to one of the cited methods, pciSeq, but the authors have provided an alternative method to achieve the same objectives. The authors need to improve the description and distinction of prior work, as well as, bench-mark their method to prior relevant methods. In the introduction they write "both pciSeq and SSAM are limited to cell type information and do not create a segmentation map for the assignment of genes that are not cell type markers" This sentence needs clarification. Both JSTA and pciSeq use cell type definition from single cell RNAseq to segment cells, and use all in situ data to do that.

While JSTA and pciSeq are conceptually similar, the key distinction is that JSTA's goal is to jointly segment and classify cells whereas pciSeq's goal is not cell segmentation, but cell type classification using spatial information. We discussed the distinctions with the authors of pciSeq and they wrote to us that "It [pciSeq] doesn't try to segment an image. The method uses the segmentation as an input to do the cell typing. We want to know which spots are placed within the boundaries and this info is taken into account when the likelihood is calculated." To further clarify this point, we now revised the text in the third paragraph of the introduction to be:

"pciSeq's primary goal is to assign cell types to nuclei by using proximity to mRNA, and an initialized segmentation map to compute the likelihood of each cell type."

SSAM creates cell type maps based on the mRNA distribution, but does not produce cell boundaries, and therefore a cell segmentation map. To further clarify this point, we now revised the text in the third paragraph in the introduction to be:

"Similarly, SSAM creates cell type maps based on RNA distributions, without creating a cell segmentation map because it ignores cellular boundaries."

1.2. It's a bit unclear what the distinction of marker genes and non-marker genes is in the results section. I assume that the marker genes are the ones that have implications in setting the cell boundaries in their model, and the non-marker genes are the ones that are "assigned" to a cell based on the segmentation map. The question is: how do you decide that a gene is a marker gene or not? Most genes are differentially expressed between cells to some extent.

We appreciate the reviewers comments pointing out the lack of clarity about marker vs non-marker genes. We completely agree that most genes will be differentially expressed between cell types. However, subsets of genes show stronger differential expression between cell types and can distinguish cell identities better than the others. For example, the 163 genes panel used here was designed to address biological questions related to traumatic brain injury. Among these, 83 genes were chosen based on prior data that they do not change after injury but are differential between cell types (defined as cell marker genes here) while 80 genes were chosen based on prior information that they show differential expression under brain injury (defined as non-marker genes). As we do not want segmentation results to depend on experimental perturbations such as injury we only used the marker genes for segmentation and we show how other genes could be assigned to cells. We further clarify that we only used a subset of the genes as cell marker genes for segmentation.

In the second paragraph of the “*JSTA overview and methods*” subsection of the results, we revised the text as follows: “We performed Multiplexed Error Robust Fluorescent In Situ Hybridization (MERFISH) of 163 genes which include 83 selected cell marker genes which show distinct expression between cell types and are used for cell classification and segmentation and 80 genes previously implicated with biological importance in traumatic brain injury.”

In the fourth paragraph of the “*JSTA overview and method*” subsection of the results, we revised the text as follows: “JSTA can be applied on any user selected subset of the genes; the local mRNA density of these selected genes around each pixel are used as the input for the pixel level classifier. The selection of genes drives how well the cell type classifier can distinguish between distinct cell types.”

1.3. An important pre-requirement for this method is the necessity of harmonizing the scRNAseq data and the spatial dataset. It would be interesting to see some simulations and discussion of potential pit-falls in this harmonization. How does variation in data density in the scRNA seq data set, and the in situ data set affect reproducibility of this harmonization?

We appreciate the reviewer’s suggestions regarding data harmonization. The full problem of data harmonization is a complex one and is an active research topic¹⁻⁴. One of the strengths of single cell analysis tools is their modularity and the ability to mix and match as needed. Here we employed a straightforward harmonization method of centering and scaling the expression level of genes within a cell and across cells each dataset. Given that harmonizing, i.e. identifying a mapping between the two data types, is a preprocessing step, as better tools are developed we anticipate that they could be used together with JSTA to create more refined analysis pipelines.

We added this point in the second paragraph of the Discussion section:

“JSTA is limited by its ability to harmonize technical differences between spatial transcriptomics data modalities and the scRNAseq reference. Harmonization between datasets is an active area of research, and JSTA will benefit from these advances.”

Additionally, we added this point to the “Cell Type Classification” section of the methods as follows:

“To match the distributions of both scRNAseq and MERFISH, we centered and scaled each cell across all genes. We then subsequently centered and scaled each gene across all cells. We note that other harmonization approaches could be applied here.”

1.4. Since the method is based on an initial watershed segmentation with a decent quality, it would be interesting to know what happens with cells where this segmentation goes wrong. Is the algorithm able to split cells incorrectly merged by watershed, or conversely, join incorrectly split cells?

Watershed is based on two steps. Initial identification of one seed per cell and the assignment of pixels into these cells. In the first step of watershed, if the seed identification is wrong, this can cause two cells to become one, or one to become two. The benefit of JSTA, as we show in our benchmarks, is that by taking cell type information into account, assignment of pixels is improved. JSTA does not perform cell splitting or merging. Previous approaches in microscopy employ splitting and merging to improve on watershed⁵⁻⁸. JSTA can be improved in a similar manner that the basic watershed algorithm can be improved by adding steps of splitting and merging. We discuss this point in the second paragraph of the discussion related to future development.

We add the following addition to the discussion section: “JSTA relies on initial seed identification (nuclei or cell centers), and incorrect identification can lead to split or merged cells. JSTA currently does not split or merge cells, but this post processing step could be added to further improve segmentation.”

1.5. When evaluating the performance of the JSTA method, it would be very useful to include a comparison to how other published segmentation methods performs. There are several out there now, in addition to pciSeq and SSAM, there is e.g. spage2vec (<https://febs.onlinelibrary.wiley.com/doi/10.1111/febs.15572>). A minimum requirement would be to benchmark with pciSeq since they are conceptually very similar.

We appreciate the reviewers suggestions to benchmark against pciSeq. We attempted to benchmark JSTA against pciSeq, Bayor, and spage2vec. We note that none of these tools worked in our hands straight from github. We were unable to use the code provided for spage2vec and Baysor and there was limited response when we contacted the authors of both packages. We are thankful for the authors of pciSeq for the tremendous help to get their code working. Using their extensive input we now add benchmark results comparing JSTA to pciSeq. We note, as discussed above in 1.1, that since the development goals of JSTA and pciSeq are different the comparison is problematic. We discuss these differences and highlight why they result in very distinct performance on different benchmarks. Additionally, pciSeq only operates on 2D data, whereas JSTA accommodates 3D data as in MERFISH. To compare the two methods, we simulated 2D samples to compare JSTA to pciSeq. Our benchmark results indicate that JSTA is much more accurate than pciSeq at assigning mRNA to cells (Figure S1a). Since pciSeq is not designed for this, it incorrectly assigns many mRNA molecules to background. If we ignore the mRNAs assigned to background and only consider the fraction of mRNA molecules assigned to cells in the true positive rate calculation, pciSeq performs well as it mainly assigns mRNA spots near the nucleus, and JSTA is comparable using this metric (Figure S1b). Overall, for the task of cell segmentation, for which it was developed, JSTA has better performance than pciSeq as JSTA assigns many more mRNAs to the correct cells and not mis-assign them to the background.

We added the following accompanying text to the “Performance evaluation using simulated hippocampus data” subsection of the results:

“We additionally compared JSTA to pciSeq⁹, in the assignment of mRNA molecules to cells. We note that pciSeq is mainly designed to assign cell types to nuclei based on surrounding mRNA, and therefore is not primarily focused on assigning most mRNA molecules to cells as JSTA does. Furthermore, since pciSeq is not designed to operate on 3D data, we simulated 2D data and applied both JSTA and pciSeq. We found that JSTA was more accurate at assigning mRNA molecules to cells than pciSeq (Figure S1a). pciSeq tends to incorrectly assign many spots to background, as segmentation is not its primary goal. However, when ignoring mRNAs assigned to background in a true positive calculation, pciSeq performs well as it primarily assigns mRNAs close to the nuclei, which is an easier task. In this case, JSTA has comparable performance (Figure S1b).”

Figure S1. Performance evaluation of JSTA, pciSeq, and watershed. pciSeq is unable to run on 3D data (solid line), so we simulated 2D data (dotted line). We evaluated the three methods on the performance of accuracy of assigning mRNAs to the correct cell (a). JSTA is more accurate than pciSeq on the accuracy metric, whereas pciSeq incorrectly assigns many mRNAs to background. We additionally tested these methods on their performance of assigning mRNAs to the correct cell while ignoring those assigned to background (b). In this case pciSeq performs better, because it mainly assigns mRNA spots close to the nucleus; JSTA has comparable performance.

1.5. Figure 4: What's the x-axis?.

We have modified Figure 4 to clarify the x-axis as the lateral-medial axis.

1.6. In the discussion, the authors mention that JSTA could be applied to other spatial technologies, and mention spatial barcoding approaches, referring to ST/Visium. I do not see how this could be done.

One of the first preprocessing steps in JSTA is density estimation that converts the data from spot-based to position-based. This makes it very comparable to ST/Visium. We agree with the reviewer that this will only work when ST/Visium provide subcellular resolution, which is achievable in the foreseeable future based on public disclosure from 10x Genomics. We clarify this point in the text by using the term **subcellular** spatial barcoding.

Reviewer #2:

2.1. The submitted manuscript MSB-2020-10108 in Molecular Systems Biology entitled 'Joint cell segmentation and cell type annotation for spatial transcriptomics' by Littmann and colleagues describes and applies their Joint cell Segmentation and cell Type Annotation (JSTA) analysis workflow to MERFISH spatial transcriptomics data of the mouse hippocampus. It combines an initial watershed segmentation algorithm with mRNA-based pixel classifiers to assign cell types for spatial transcriptomics, whereas JSTA derives pixel probabilities from deep neural network cell and pixel classifiers. By spatial differential gene expression segmented mouse hippocampus cells are further stratified using non-cell type markers. Although the JSTA method seems to be conceptually useful, any comparison is missing.

As discussed in the response to reviewer comment 1.5 above, we now add a detailed benchmark of JSTA against pciSeq.

Major concerns:

2.2. The manuscript is based on only one data set (MERFISH). It would be recommended if Littmann et al. evaluate JSTA also one other data set to proof the concept.

Baysor (<https://www.biorxiv.org/content/10.1101/2020.10.05.326777v1>) and SSAM use e.g. osmFISH and MERFISH. Please introduce Baysor in the intro as well. Different alternative other approaches are also mentioned in the discussion as applications (ISS and spatial barcoding).

We appreciate the reviewer's suggestions to add Baysor to the introduction and apply JSTA to additional datasets. Baysor has been added to the third paragraph of the introduction with the following text: "More recently, an approach for updating cell boundaries in spatial transcriptomics data has been developed. Baysor uses neighborhood composition vectors and markov random fields to segment spatial transcriptomics data and identify cell type clusters."

We also applied JSTA to two additional datasets: a MERFISH dataset of the mouse hypothalamic preoptic region¹⁰ and an osmFISH dataset of the mouse somatosensory cortex¹¹. We find that JSTA accurately maps cell types and segments cells for both datasets.

For the MERFISH dataset of the mouse hypothalamic preoptic region¹⁰, we used the 134 genes provided, and leveraged their scRNAseq dataset to map 87 high resolution cell types to this region. The scRNAseq dataset published with this paper was used as the reference dataset. Our findings of the spatial distribution of high resolution cell types were consistent with the manually annotated findings in the original manuscript (Figure S2a). We additionally highlight that the average gene expression profile of cell types in the scRNAseq are highly correlated with those annotated by JSTA using the MERFISH data (Figure S2b). Full correlation matrix with high resolution cell type labels is provided (supplementary table 3). We added the following associated section to the results:

JSTA performs high resolution cell type mapping in the mouse hypothalamic preoptic region

We applied JSTA to a MERFISH from a previously published mouse hypothalamic preoptic region with 134 genes provided¹⁰. Using the provided scRNAseq reference dataset, we accurately mapped 87 high resolution cell types in this region (Figure S2a). The mapped cell types follow spatial distributions of high resolution cell types of this region previously annotated through clustering and marker gene annotation. We find the gene expression profiles of the cell types from the MERFISH data is highly correlated with their scRNAseq counterparts (Figure S2b).

For the osmFISH dataset of the somatosensory cortex (Codeluppi et al. 2018), we used the 35 genes provided, and leveraged the NCTT reference to map the 142 high resolution cell types of the somatosensory cortex. The glutamatergic cell types were specifically mapped to their known layers in the cortical region by JSTA (Figure S3a). We also found that the gene expression patterns of the high resolution cell types in scRNAseq data are highly correlated with those mapped by JSTA in the osmFISH dataset (Figure S3b). Full correlation matrix with high resolution cell type labels is provided (supplementary doc#). We added the following associated section to the results:

JSTA performs high resolution cell type mapping in the mouse somatosensory cortex

Next, we applied JSTA to an osmFISH dataset from the mouse somatosensory cortex with the 35 genes provided. Using the NCTT reference, we mapped 142 high resolution cell types in this region. We found that the glutamatergic neuronal populations follow known spatial organization (Figure S3a), and that the gene expression patterns of high resolution cell types in the osmFISH data are highly correlated with their NCTT counterparts (Figure S3b).

These results provide evidence that JSTA can accurately learn cell type representations from different spatial transcriptomics datasets as well as different scRNAseq references.

These results are now shown in Supp Figures 2 and 3 below.

a)

b)

Figure S2. Application of JSTA to MERFISH data from the mouse hypothalamic preoptic region. High resolution cell types identified by JSTA (a). The spatial mappings of these high resolution cell types are consistent with the manually annotated data from Moffit et al. JSTA mapped high resolution (sub)types are highly correlated to their scRNAseq reference counterparts in terms of gene expression patterns (b; Supplementary table 3). Cell types with at least 5 cells were kept.

Figure S3. Application of JSTA to osmFISH data from the mouse somatosensory cortex. Glutamatergic neurons are consistent with previously identified spatial patterns of the somatosensory cortex (a). JSTA mapped high resolution (sub)types are correlated with their NCTT counterparts in terms of gene expression patterns (b; Supplementary table 4). Cell types with at least 5 cells were kept.

2.3. The intro about other analysis pipelines is at least misleading. Their introduction: "both pciSeq and SSAM are limited to cell type information" is not true as modelling non-marker genes is possible, but not shown. ", and do not create a segmentation map for the assignment of genes that are not cell type markers". This is unfair to pciSeq, as this assigns cell-types to cells (based on segmentation). Please carefully rewrite the whole paragraph similar to 'pciSeq assigns cell types to prior segmented regions based on proximity to mRNA of marker genes, circumventing the need for pixel level segmentation. Similarly, SSAM creates cell type maps based on RNA distributions without needing to know cellular boundaries. However, both pciSeq and SSAM do not create a segmentation map. Therefore, while both pciSeq and SSAM leverage cell type catalogs to provide insights into the spatial distribution of different cell types they do not produce a cell segmentation map and are only demonstrated on cell (sub)type label information (cell type map).' Again, the description of not being able to consider non-marker genes is inaccurate. All this might change the impact/uniqueness of the manuscript (including Baysor, Petukhov et al.).

We thank the reviewer for this comment. Very similar comments was pointed out by reviewer #1. We discussed our revisions in more detail in 1.1 In brief, clarify the distinction between pciSeq, SSAM and JSTA. We explain the primary goal of pciSeq is for cell type identification, and SSAM creates a cell type map, but does not segment cells by identifying cell borders. The revised text is as follows: "pciSeq's primary goal is to assign cell types to nuclei by using proximity to mRNA, and an initialized segmentation map to compute the likelihood of each cell type. Similarly, SSAM creates cell type maps based on RNA distributions, without creating a cell segmentation map because it ignores cellular boundaries. Therefore while both pciSeq and SSAM leverage cell type catalogs to provide insights into the spatial distribution of different cell types they do not produce a high quality cell segmentation map."

2.4. Consequently, the spDEG of JSTA in Fig. 6 (SFig. 3) with 50% of tested genes showing a non-random distribution is probably not an exclusive feature of JSTA. It is unclear how other methods mentioned in the introduction or application would perform here.

Indeed, we believe the non-random distribution of many genes is more of a biological property of those genes than being the technical capacity of JSTA. Other methods can retrieve similar patterns as long as the cell/gene assignments are accurate. The identification of spDEG is not a feature of JSTA but a straightforward analysis one can do with good cell segmentations and RNA assignment that we used to demonstrate the application of JSTA. We add this point at the end of the "*JSTA identifies spatial differential gene expression (spDEGs)*" subsection of the results: "Identification of spDEGs highlights an interesting application of highly accurate cell type and mRNA assignment in spatial transcriptomics data."

Minor concern:

2.5. Please add page numbers.

We have added page numbers.

2.6. Please revise axis labels Fig. 3d? Maybe Fig. 3 subfigure without any JSTA segmentation in correlation with NCTT would be useful to see the advances.

The goal of figure 3 is to indicate that the cell type identification performed by JSTA is accurate, by showing on average the cell types are correlated with their scRNAseq counterparts. Since the correlation is computed using the average expression of each cell type, we do not expect JSTA segmentation to largely change the

correlation structure here. We are unsure what the reviewer is asking regarding the axis labels of Figure 3d and have kept them as they are as the goal of this panel is to show the correlation between major cell types in the scRNAseq data is comparable to what we recover with MERFISH (panel 3c)

2.7. Overall labels, abbreviations and headers in figure legends should be more detailed.

We have thoroughly revised our figure legends to provide detailed explanation of the labels, abbreviations, and headers.

We have modified the figure legend for Figure 1:

Figure 1. Overview of JSTA and the spatial transcriptomics data used for performance evaluation. **a.** Joint cell segmentation and cell type annotation (JSTA) overview. Initially, watershed based segmentation is performed and a cell level type classifier is trained based on the Neocortical Cell Type Taxonomy (NCTT) data. The deep neural network (DNN) parameterized cell level classifier then assigns cell (sub)types (red and blue in this cartoon example). Based on the current assignment of pixels to cell (sub)types, a new DNN is trained to estimate the probabilities that each pixel comes from each of the possible (sub)types given the local RNA density at each pixel. In this example, two pixels that were initially assigned to the “red” cells got higher probability to be of a blue type. Since the neighbor cell is of type “blue” they were reassigned to that cell during segmentation update. Using the updated segmentation and the cell type classifier cell types are reassigned. The tasks of training, segmentation, and classification are repeated over many iterations until convergence. **b.** Multiplexed Error Robust Fluorescent in situ hybridization (MERFISH) and DAPI stained nuclei in the mouse hippocampus. Each gene is represented by a different color. For the entire hippocampus (left), only the mRNA spots are shown with a scale bar of 500 microns. On the zoomed-in section (right), each gene is represented by a different color dot, and the DAPI intensity is displayed in white. The scale bar is 20 microns.

We have modified the figure legend for Figure 2:

Figure 2 Performance evaluation of JSTA using simulated data. **a.** Representative synthetic dataset of nuclei (black) and mRNAs, where each color represents a different gene. **b.** Ground truth segmentation map of the cells in the representative synthetic dataset. Each color represents a different cell. **c.** Average Accuracy of calling mRNA spots to cells at different cell type resolutions using 83 genes across 10 replicates. Accuracy was determined by the assignment of each mRNA molecule to the correct cell type. JSTA (solid line) is more accurate than watershed (dashed line) at assigning mRNA molecules to the correct cells (FDR < 0.05). Statistical significance was determined with a Mann-Whitney test and false discovery rate correction. **d.** Accuracy (as described in c) of calling mRNA spots to cells when using JSTA to segment cells with a lower selection of cell type marker genes (8-44 genes tested). The color of the line gets progressively darker as the number of genes used increases.

We have modified the figure legend for Figure 3:

Figure 3 Segmentation of MERFISH data from the hippocampus using JSTA. **a.** High resolution cell type map of 133 cell (sub)types segmented and annotated by JSTA. Colors match those defined by Neocortical Cell Type Taxonomy (NCTT). Scale bar is 500 microns. **b.** JSTA based classification of CA1 (green), CA3 (cyan), and DG (red) neurons matches their known domains. **c.** Correlation of the average expression of 163 genes across major cell types between MERFISH measurements to scRNAseq data from NCTT. **d.** Correlation of the average expression of the same genes as in c between expression of types in scRNAseq data from NCTT. The correlation structure in c closely mirrors the structure in d.

We have modified the figure legend for Figure 4:

Figure 4 Spatial distribution of neuronal subtypes in the hippocampus. **a(i).** Cell subtype map of CA1 neurons in the hippocampus as annotated by JSTA. Scale bar is 500 micron. Distribution of CA1 subtypes in

the hippocampus, computed by projecting cell centers to the lateral to medial axis. CA1 neuronal subtypes show a non-uniform distribution across the whole CA1 region. **a(ii)** Smoothed histogram highlighting the density of CA1 subtypes across the CA1 region. **b(i)**. Cell subtype map of CA3 neurons in the hippocampus as annotated by JSTA. Distribution of CA3 subtypes in the hippocampus, computed by projecting the cell centers to the lateral to medial axis. CA3 neuronal subtypes show a non-uniform distribution across the whole CA3 region. **b(ii)** Smoothed histogram highlighting the density of CA3 subtypes across the CA3 region.

We have modified the figure legend for Figure 5:

Figure 5 Agreement between spatial proximity and gene coexpression in highly granular cell subtypes in the hippocampus. **a-b.** Relationship between the frequency of a (sub)type's neighbors and its transcriptional pearson correlation between CA1 subtypes (**a**) and between CA3 subtypes (**b**). **c.** Cell type map in the hippocampus shows specific colocalization patterns between a subset of Sst subtypes (purple) and CA1 neurons (green); these Sst subtypes do not colocalize with CA3 neurons (cyan). **d.** Colocalization patterns of Sst subtypes with CA1 and CA3 subtypes. Sst subtypes that colocalize with the CA1 subtypes have high transcriptional similarity. Colocalization was defined as the percent of neighbors that are of that subtype (methods) **e** Transcriptional correlation patterns between Sst subtypes and CA1 and CA3 neurons. Green, purple and cyan sidebars highlight the subset of Sst co-localized with CA1 (purple), CA1 (green) and CA3 (cyan).

We have modified the figure legend for Figure 6:

Figure 6 Identification of spatial differential gene expression (spDEGs). **a.** Normalized expression of *Tox* in CA1sp1, *Leng8* in CA3sp3, *Hecw1* in DG3, and *Thra* in Astro1 show variable expression throughout the hippocampus. Scale bar is 500 microns. spDEGs were computed by comparing the true variance in gene expression between cell subtype neighborhoods to that of randomly permuted cell (sub)type neighborhoods. **b.** Histogram of the number of statistically significant spDEGs (Benjamini-Hochberg corrected FDR < 0.05) in each subtype. **c.** Histogram of the number of subtypes that have an spDEG for each gene.

We have modified the figure legend for Figure S4:

Figure S4. Cross entropy loss and accuracy of cell type (**a-b**) and pixel (**c-d**) classifier during training for the train (blue) and validation (orange) data sets. **a-b.** Cross entropy (**a**) loss and accuracy (**b**) during training cell type classifier. The cell type classifier overfits the training data, and is mitigated by stopping training after 40 epochs. **c-d.** Cross entropy loss (**c**) and Accuracy (**d**) during training of the pixel classifier. Black lines indicate new training iteration after pixel reassignment.

We have modified the figure legend for Figure S5:

Figure S5. Correlation structure of cell types compared to their colocalization. Neuronal subtypes that are highly colocalized are often correlated in their gene expression. Cell types with more than 10 cells were included. **a.** pearson correlation of 122 (sub)types across 83 selected genes. **b.** Frequency of neighbors between each of 122 (sub)types. Only significant (FDR < 0.05) colocalizations are shown. Labels and values are detailed in supplementary table 5 and 6.

We have modified the figure legend for Figure S6:

Figure S6. Identification of spatial differentially expressed genes (spDEGs). spDEGs were computed by comparing the true variance in gene expression between cell subtype neighborhoods to that of randomly permuted cell (sub)type neighborhoods **a.** 63 genes across 61 cell types show significant spDEGs. Heatmap values correspond to $-\log_2(p\text{-value})$. **b.** Number of spDEGs in each of the 61 cell types. **c.** Number of cell types with each of the 63 spDEGs.

2.8. Overall, the manuscript leaves the reader with the question: how significant are these technical advances in comparison?

See discussion on our newly added benchmarking results in 1.5 and additional application examples in 2.2 above, which demonstrate the performance and broad utility of JSTA.

Reviewer #3:

This paper presents an optimization approach for segmentation and annotation of single cells from spatial transcriptomics datasets. Unlike conventional approach using watershed, the authors propose a pixel based density and optimization approach for classification of transcript data to associate cell subtypes based on the known expression of individual cells.

The JSTA approach is interesting, however, the following can be addressed to improve the paper:

3.1. Nowadays the gold standard for cell segmentation and annotation is to also perform protein staining on the same samples after the MERFISH experiments. The cell boundaries using FISH labels can be misleading as the density of RNA molecules is highly variable. Authors should validate their segmentation results by comparing the data to a cell mask that is generated from a surface protein staining.

The goal of our study is to develop a new method to segment cells and annotate genes to cell types for spatial transcriptomics data. The accuracy is established through in silico validation and comparison with other existing methods. Protein assays are not used in any of the previous cell segmentation method studies^{9,12,13}.

3.2. The number of transcripts per pixel would change based on the biological significance of each gene for unique cell types. Would this create bias in the annotation and segmentation of the JSTA approach?

We first center and scale the gene expression within each cell making the annotation in JSTA based on relative expression of genes within each cell. The number of transcripts per pixel should not have an effect on the annotation. Since JSTA uses a two classifier system, where the second (pixel) classifier learns the distribution of genes in pixels for each cell type, and the annotation would not be affected by the number of transcripts per pixel, the overall segmentation should not be affected.

3.3. The pixel distributions of each pixels would be beyond 3x3 pixels due to diffraction limit and overlapping RNA molecules would be also making the assignment of specific RNA type complicated. Could authors change the pixel window from 3x3,5x5, and 7x7 etc. to fine tune their results that can potentially reduce the effect of diffraction limit.

Our KDE is bigger than the diffraction limit, so the effect of the diffraction limit is minimal when changing the pixel window to bigger than 3x3.

3.4. How would this approach compare to a deep learning framework (<https://www.nature.com/articles/s41592-019-0403-1>) that can be applied to either pixel data or cells directly? Would a different model e.g., Keras improve these segmentation results?

We thank the reviewer for this suggestion. We already use cellpose, a deep learning method to segment nuclei. The challenge is that outside of nuclei there is no information about cell identity. JSTA leverages

external scRNAseq data and prior knowledge of cell types to address this issue. As for the question related to Keras, JSTA incorporates deep learning tools available in Keras.

3.5. Which dyes were used for these MERFISH experiments. Is it only Alexa 647 or multiple colors? If so, how would the authors address the pixel level shifts of transcripts due to the optical aberrations?

We use Cy5, Atto 565, Alexa 488, and Hoechst. Following Moffitt et al.¹⁰ we perform chromatic aberration correction. We note that these steps are important for MERFISH for spot calling accuracy; they will have no impact on the performance of JSTA given spatial gene expression data.

3.6. How would this JSTA scale up with data size? If the entire mouse brain were imaged, would it be computationally efficient to run these classifiers in a reasonable time. In other words, how long does it take and at which computation speed?

We thank the reviewer for their comments regarding the computational time for JSTA. We simulated 2D data with 6 different sizes, and replicates per group, and noted the computational time for JSTA. We find that JSTA scales linearly with both the area of the tissue section, and the number of cells in the section. In half of a coronal section of the mouse brain, the sample imaged in Ortiz et al. was about 24 million square microns¹⁴. In this scenario we expect JSTA to run in about 100 minutes, but could also be sped up if the slide were divided into parts or with the use of GPU/TPU accelerators. We added this analysis as Figure S7 below, and as a section in the results:

Time requirements of JSTA

We simulated data of different sizes and ran JSTA to determine how the run time scales with larger datasets. We simulated 3 replicates of data with a width and height of 100, 200, 300, 400, 500, and 1000 microns. The run time of JSTA scales linearly with both the area and number of cells in the section (Figure S7ab).

Figure S7. Run time evaluation of JSTA on simulated data. We ran JSTA on data simulated with a width and height of 100, 200, 300, 400, 500, and 1000 microns, with 3 replicates each. We evaluated the time taken to run JSTA by the area of the section (a), and the number of cells in each section (b).

3.7. Finally, while they are not the same, how would this approach complement GIOTTO

(<https://www.biorxiv.org/content/10.1101/701680v1.full>) and TANGRAM

(<https://www.biorxiv.org/content/10.1101/2020.08.29.272831v3>)? What is the advantage of the JSTA compared to the recent emerging literature including these two recent platforms?

The ecosystem of single cell and spatial transcriptomics analysis is extensive and growing rapidly. There are many tools that address different analysis tasks. GIOTTO and TANGRAM are two of many such tools. To avoid confusion we focus on discussing only relevant tools that solve the same problem that JSTA does: cell segmentation and type assignment. JSTA is complementary to GIOTTO, as the authors state: “Cell boundary segmentation is a necessary step for assignment of each transcript to its corresponding cells. However, this task is highly dependent on the specific technology platform therefore not implemented in Giotto”. TANGRAM similarly uses segmented spatial transcriptomics data as an input¹⁵. Therefore, JSTA offers output that can be used in GIOTTO and TANGRAM. We have added the following comment to the Discussion:

“Cell segmentation results from JSTA can be used as input for other tools such as GIOTTO¹⁶ and TANGRAM¹⁵ to facilitate single cell and spatial transcriptomic data analysis.”

3.8. The cell segmentation and annotation issues are primary concerns of cell atlas studies that can impact the understanding of basic mouse biology and human health. These solutions will be highly important to enable reproducible and efficient data analysis pipelines for spatial transcriptomics technologies.

We agree and have emphasized the significance of such methods and the importance of reproducibility in our discussion.

References

1. Tran, H. T. N. *et al.* A benchmark of batch-effect correction methods for single-cell RNA sequencing data. *Genome Biol.* **21**, 12 (2020).
2. Korsunsky, I. *et al.* Fast, sensitive and accurate integration of single-cell data with Harmony. *Nat. Methods* **16**, 1289–1296 (2019).
3. Welch, J. *et al.* Integrative inference of brain cell similarities and differences from single-cell genomics. doi:10.1101/459891.
4. Stuart, T. *et al.* Comprehensive Integration of Single-Cell Data. *Cell* **177**, 1888–1902.e21 (2019).
5. Gamarra, M., Zurek, E., Escalante, H. J., Hurtado, L. & San-Juan-Vergara, H. Split and merge watershed: A two-step method for cell segmentation in fluorescence microscopy images. *Biomed. Signal Process. Control* **53**, 101575 (2019).

6. Surut, Y. & Phukpattaranont, P. Overlapping cell image segmentation using surface splitting and surface merging algorithms. in *Second APSIPA Annual Summit and Conference* 662–666 (2010).
7. Correa-Tome, F. E. & Sanchez-Yanez, R. E. Integral split-and-merge methodology for real-time image segmentation. *JEI* **24**, 013007 (2015).
8. Chaudhuri, D. & Agrawal, A. Split-and-merge procedure for image segmentation using bimodality detection approach. *Def. Sci. J.* **60**, 290–301 (2010).
9. Qian, X. *et al.* Probabilistic cell typing enables fine mapping of closely related cell types in situ. *Nat. Methods* **17**, 101–106 (2020).
10. Moffitt, J. R. *et al.* Molecular, spatial, and functional single-cell profiling of the hypothalamic preoptic region. *Science* **362**, (2018).
11. Codeluppi, S. *et al.* Spatial organization of the somatosensory cortex revealed by osmFISH. *Nature Methods* vol. 15 932–935 (2018).
12. Petukhov, V., Soldatov, R. A., Khodosevich, K. & Kharchenko, P. V. Bayesian segmentation of spatially resolved transcriptomics data. *Cold Spring Harbor Laboratory* 2020.10.05.326777 (2020) doi:10.1101/2020.10.05.326777.
13. Park, J. *et al.* Cell segmentation-free inference of cell types from in situ transcriptomics data. *Cold Spring Harbor Laboratory* 800748 (2020) doi:10.1101/800748.
14. Ortiz, C. *et al.* Molecular atlas of the adult mouse brain. *Sci Adv* **6**, eabb3446 (2020).
15. Biancalani, T., Scalia, G., Buffoni, L., Avasthi, R. & Lu, Z. Deep learning and alignment of spatially-resolved whole transcriptomes of single cells in the mouse brain with Tangram. *bioRxiv* (2020).
16. Dries, R. *et al.* Giotto, a pipeline for integrative analysis and visualization of single-cell spatial transcriptomic data. *Cold Spring Harbor Laboratory* 701680 (2019) doi:10.1101/701680.

Thank you for sending us your revised manuscript. We have now heard back from the three reviewers who were asked to evaluate your study. Overall, the reviewers think that the study has improved as a result of the performed revisions. As you will see below, reviewer #2 still has some remaining concerns, regarding the benchmarking. As it does not seem that reviewer #2 is opposed to the publication of the study despite these concerns, and given that reviewers #1 and #3 are supportive, we would ask you to address the remaining concerns of reviewer #2 in a minor revision by performing some text changes and discussing these potential limitations.

When you revise your manuscript, we would also ask you to address some editorial issues listed below.

REFeree REPORTS

Reviewer #1:

The manuscript has improved substantially, and I think it is now acceptable for publication.

I still think there is some confusion with respect to the meaning of the word "segmentation". I believe reviewers 1 and 2 have a different opinion compared to the authors about what segmentation is and what it does. I do believe that with the revision, at least it is now clearer how the authors use it, and thereby a lot of the confusion has been resolved.

Reviewer #2:

In principle the concerns of Reviewer #1 and #2 are indeed very similar. They refer to other tools which should be used in comparison (pciSeq, Bayor, space2vec, SSAM). Littman et al. state that they could not properly use Bayor and space2vec because the code 'did not work straight from GitHub' in their hands and they received 'limited response' from the authors. This is probably true, but usually one has to prove better performance in comparison to existing algorithms and only isolated measurements from raw data is not helpful for the community. In this revision, at least pciSeq was tested in comparison (SFig. 1), which shows JSTA better in an accuracy 'metric' and pciSeq better in the true positive rate. SSAM was not even further mentioned. This means this revision did not sufficiently address the 'benchmarking' concerns. Finally, all these methods might perform better than the labs developing Spatial Transcriptomics (MERFISH or additional osmFISH), but which algorithm provide most biological insights is measured by cell types and genes, not by segmented cells, a stated unique feature of the algorithm (Reviewer point 3.4 about segmentation is valid and also not met). I hope that the other preprint revisions will be treated accordingly.

Reviewer #3:

The revision has addressed the questions raised by the referee. Thus, I recommend publication of this paper.

We thank reviewer #2 continued discussion about the role of segmentation in spatial transcriptomics data. To address a few of the remaining points:

In principle the concerns of Reviewer #1 and #2 are indeed very similar. They refer to other tools which should be used in comparison (pciSeq, Bayor, space2vec, SSAM). Littman et al. state that they could not properly use Bayor and space2vec because the code 'did not work straight from GitHub' in their hands and they received 'limited response' from the authors. This is probably true, but usually one has to prove better performance in comparison to existing algorithms and only isolated measurements from raw data is not helpful for the community. In this revision, at least pciSeq was tested in comparison (SFig. 1), which shows JSTA better in an accuracy 'metric' and pciSeq better in the true positive rate. SSAM was not even further mentioned. This means this revision did not sufficiently address the 'benchmarking' concerns.

We did whatever benchmarking we could in the previous revision. SSAM is explicitly not a cell segmentation tool. In fact, the title of the preprint describing this method is: "Cell segmentation-free inference of cell types from in situ transcriptomics data". Therefore, by definition, we can't benchmark JSTA against SSAM as this is a clear case of apples and oranges.

Finally, all these methods might perform better than the labs developing Spatial Transcriptomics (MERFISH or additional osmFISH), but which algorithm provide most biological insights is measured by cell types and genes, not by segmented cells, a stated unique feature of the algorithm

I am not sure what point the reviewer is trying to make here. JSTA is a tool for cell segmentation and type assignment for spatial transcriptomics. We are not arguing that cell segmentation is required to answer ALL possible biological questions that one could have. Rather, if the biological question necessitates cell segmentation, a step that is currently done in the majority of spatial transcriptomics papers, JSTA is a powerful and useful tool.

(Reviewer point 3.4 about segmentation is valid and also not met). I hope that the other preprint revisions will be treated accordingly.

We addressed that point in the previous revision, it is unclear to us why the reviewer considers this as an "unmet concern".

Thank you again for sending us your revised manuscript and for performing the requested changes. We are now satisfied with the modifications made and I am pleased to inform you that your paper has been accepted for publication.

Corresponding Author Name: Roy Wollman

Manuscript Number: MSB-2020-10108